# SpidR: Learning Fast and Stable Linguistic Units for Spoken Language Models Without Supervision

**Maxime Poli**[1,†], **Mahi Luthra**[2,*], **Youssef Benchekroun**[2,*], **Yosuke Higuchi**[2], **Martin Gleize**[2], **Jiayi Shen**[2], **Robin Algayres**[2], **Yu-An Chung**[2], **Mido Assran**[2], **Juan Pino**[2], **Emmanuel Dupoux**[1,2]

[1] *ENS-PSL, EHESS, CNRS,* [2] *FAIR at Meta*                                     *maxime.poli@ens.psl.eu*

[†] work done in part while interning at Meta
[*] equal contribution

**Reviewed on OpenReview:** https://openreview.net/forum?id=E7XAFBpfZs

## Abstract

The parallel advances in language modeling and speech representation learning have raised the prospect of learning language directly from speech without textual intermediates. This requires extracting semantic representations directly from speech. Our contributions are threefold. First, we introduce SpidR, a self-supervised speech representation model that efficiently learns representations with highly accessible phonetic information, which makes it particularly suited for textless spoken language modeling. It is trained on raw waveforms using a masked prediction objective combined with self-distillation and online clustering. The intermediate layers of the student model learn to predict assignments derived from the teacher's intermediate layers. This learning objective stabilizes the online clustering procedure compared to previous approaches, resulting in higher quality codebooks. SpidR outperforms wav2vec 2.0, HuBERT, WavLM, and DinoSR on downstream language modeling benchmarks (sWUGGY, sBLIMP, tSC). Second, we systematically evaluate across models and layers the correlation between speech unit quality (ABX, PNMI) and language modeling performance, validating these metrics as reliable proxies. Finally, SpidR significantly reduces pretraining time compared to HuBERT, requiring only one day of pretraining on 16 GPUs, instead of a week. This speedup is enabled by the pretraining method and an efficient codebase, which allows faster iteration and easier experimentation. We open-source the training code and model checkpoints at https://github.com/facebookresearch/spidr.

## 1 Introduction

Recent progress in self-supervised learning (SSL) (Mohamed et al., 2022) has opened up the intriguing possibility of learning language models like children do, i.e., directly from audio signals, without any text. This approach substitutes the standard text tokenizer with an SSL speech tokenizer, and trains the language model using next token prediction on speech tokens (Lakhotia et al., 2021; Dunbar et al., 2021). This opens up interesting possibilities for *textless* NLP systems that could address the large number of human languages that do not have sufficient textual resources to allow for the standard text-based or ASR-based approach (Polyak et al., 2021; Kharitonov et al., 2022; Kreuk et al., 2022; Nguyen et al., 2023; Borsos et al., 2023). In this work, we use the term *spoken language model* (SLM) to refer to language models that are trained without any text. Other authors have used this term in a broad range of situations that typically mix speech and text. We use it to refer to the *pure* spoken language models of the taxonomy proposed by Arora et al. (2025).

Early research on SLM (Lakhotia et al., 2021; Dunbar et al., 2021) used speech SSL models trained with a predictive objective (van den Oord et al., 2019; Hsu et al., 2021) as speech tokenizers. Since then, SSL models have grown in diversity and coverage, revolutionizing many aspects of speech processing for a large variety of downstream tasks (few-shot ASR, few-shot speech classification, speech compression, etc.) (wen Yang et al.,

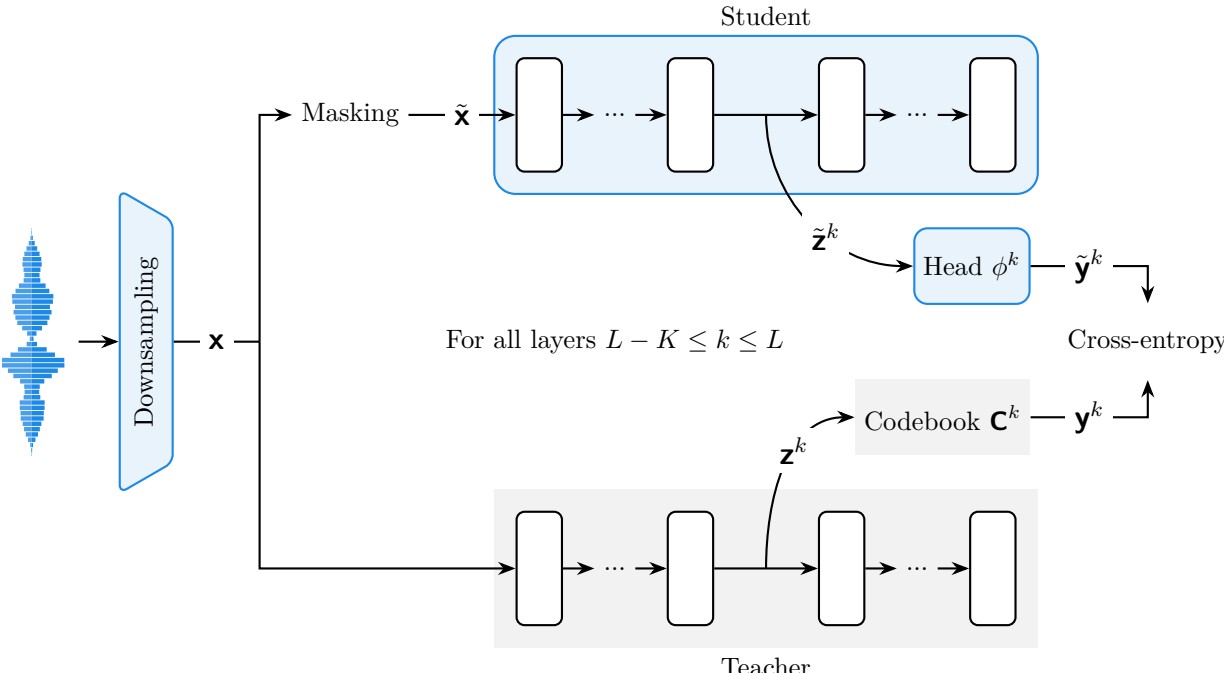

Figure 1: Architecture of SpidR. The downsampling module, a stack of convolutional layers, transforms the speech waveform into 20ms frames. The student and teacher are Transformers with $L = 12$ layers. For every layer $k$ in the last $K = 8$ ones, the student predicts—through a prediction head $\phi^k$—the nearest neighbor codebook assignment on the masked positions from the teacher at the same layer. The downsampling module, the student and the prediction heads are updated by gradient descent (in blue). The teacher is an exponential moving average (EMA) of the student, and the codebooks are updated with an EMA of the embeddings of the teacher (in gray).

2021). Yet, progress on SSL models for downstream SLM has been slow. We still have little understanding of what constitutes good speech units, and current SLMs still lag behind their textual counterpart in terms of performance and scaling laws (Cuervo & Marxer, 2024).

One popular hypothesis regarding speech units for SLM is that they should be abstract, in the sense of removing non-linguistic information like speaker identity or expressive variation, to be close to existing linguistic units like phonemes. This is why they are typically evaluated by phoneme classification metrics like ABX discriminability (Schatz et al., 2013; Schatz, 2016) or PNMI (Hsu et al., 2021). Existing speech units capture phonetic information (Choi et al., 2024), but they lack robustness to both acoustic variations (Gat et al., 2023) and contextual variations caused by coarticulation (Hallap et al., 2023). As a result, these units align more closely with contextual phone states (Young et al., 1994) than with actual linguistic units (Dunbar et al., 2022). To address these limitations, some approaches further process units derived from SSL models to make them larger and closer to syllables or words (Algayres et al., 2023; Baade et al., 2025; Visser & Kamper, 2025). However, training new SSL models from scratch is costly. For example, HuBERT is trained in several passes, and requires alternating between model training and clustering, with some manual decisions to be made between each pass to select the intermediate layer used to compute the targets. DinoSR (Liu et al., 2023a) is a recent single-pass alternative to HuBERT that demonstrates better phonetic discriminability, making it a suitable candidate for spoken language modeling. Its original implementation still requires a week of training, which limits the exploration of its training properties.

The contribution of this work has three aspects: technical, scientific, and methodological. On the technical front, we provide a minimal pure PyTorch codebase for training speech SSL models that accelerates training by an order of magnitude: full pretraining on LibriSpeech now requires only one day on 16 GPUs. This

contribution should unlock research on speech SSL models by enabling faster iterations and easier experimentation. Scientifically, we leverage this accelerated codebase to develop SpidR, a new architecture that learns strong representations for spoken language modeling in a single pass. While inspired by DinoSR's architecture, SpidR's learning objective makes pretraining significantly more resistant to codebook collapse. Our approach incorporates self-distillation and online clustering with pseudo-labels derived from codebooks at the intermediate layers of the teacher encoder. However, it differs from DinoSR in a key way: instead of using only the student's final layer to predict the assignments for each teacher intermediate layer, we use the student's own intermediate representations. Our experimental results demonstrate that SpidR outperforms both HuBERT and DinoSR on zero-shot spoken language modeling metrics. Finally, on a methodological note, we systematically evaluate across models and layers the correlation between speech unit quality (ABX, PNMI) and downstream LM performance at several levels (lexical, grammatical, and semantic). We find that unit quality strongly predicts downstream scores, validating these metrics as reliable proxies for SLM performance that enable rapid model development.

## 2 Related Work

**Self-supervised speech representation learning.** Modern self-supervised learning for speech emerged, on one hand, from the desire to leverage large amounts of unlabeled speech data to learn representations that could then be fine-tuned on smaller labeled datasets for a variety of downstream tasks, mainly ASR. On the other hand, it also follows a long history of approaches for unsupervised pattern discovery in speech (Park & Glass, 2008; Jansen et al., 2010; Kamper et al., 2015; 2016; Dunbar et al., 2022). Current methods have evolved from early autoregressive models (van den Oord et al., 2019; Schneider et al., 2019; Chung & Glass, 2020) to predominantly bidirectional masked prediction approaches (Devlin et al., 2019) that leverage surrounding unmasked context. The wav2vec 2.0 (Baevski et al., 2020) model is trained with contrastive learning between the contextual representations and quantized units. Its architecture also established a standard backbone that has been adopted by most subsequent approaches, the key differentiation between models lying in how they compute the self-supervised loss and derive training targets. HuBERT (Hsu et al., 2021) introduced an iterative approach where pseudo-targets are obtained from a previous iteration of the model, alternating between clustering and pretraining. Baevski et al. (2022) use self-distillation to derive the targets, an approach followed by DinoSR (Liu et al., 2023a) but with discrete targets instead of continuous embeddings. In self-distillation, the teacher and the student architectures are identical, and the teacher is usually an exponential moving average of the student (Grill et al., 2020; Caron et al., 2021). Using discrete targets is also the dominant approach in SSL for speech, employed by all models cited above except data2vec, as well as other architectures such as BEST-RQ (Chiu et al., 2022) and w2v-BERT (Chung et al., 2021). Although designed primarily for ASR, these models have also been repurposed for spoken language modeling: their representations capture linguistic content, allowing them to serve as discrete speech tokens, a usage more rooted in the unsupervised pattern discovery tradition. Our work builds directly on DinoSR, maintaining the established architecture while focusing specifically on improving the stability of the learning objective rather than architectural innovations. The pseudo-targets in data2vec, DinoSR and our work are derived from intermediate layers, an approach also explored by Chung et al. (2021); Wang et al. (2022). Another line of research has focused on enhancing robustness through additional training objectives—addressing acoustic or speaker variations, with approaches like WavLM (Chen et al., 2022), Spin, or R-Spin (Chang et al., 2023; Chang & Glass, 2024). Our target application in this work is not ASR or other supervised downstream tasks, but spoken language modeling, which requires different properties from representations. Word or phoneme information should be directly accessible from them, without any additional training. Particularly relevant to our goals, Chang et al. (2025) fine-tune HuBERT to learn codebooks optimized for spoken language modeling. We focus in this work on single-pass pretraining without additional fine-tuning steps, making our approach complementary to these specialized adaptation methods.

**Efficient speech representation learning.** With the increasing cost to train self-supervised speech models, researchers have explored various approaches to simplify the training procedure and accelerate training time. For instance, Baevski et al. (2023) improve the sample efficiency of data2vec by training with multiple masked versions of the same sample. For HuBERT specifically, several efficiency improvements have been proposed: Lin et al. (2023a) and Yang et al. (2023) replace the learned downsampling module

by mel-filterbanks and use a cross-entropy loss, while Chen et al. (2023a) use an existing ASR model to extract the targets for the first training iteration instead of MFCC features. Yang et al. (2025) take a different approach by replacing the encoder with a Zipformer (Yao et al., 2024). It's worth noting that training HuBERT also requires extracting features and training K-means between each iteration, which can consume a substantial portion of the total training time (Zanon Boito et al., 2024)—a inherent limitation that architectural changes alone cannot address. Additionally, most models derived from wav2vec 2.0, including HuBERT, were originally pretrained using the fairseq library (Ott et al., 2019). While fairseq initially provided essential solutions for distributed training, mixed precision, etc., these features now exist natively in PyTorch (Ansel et al., 2024), and fairseq is no longer maintained. Our streamlined PyTorch-native implementation of SpidR and DinoSR reduces compute requirements, enables faster iteration during development, and provides a hackable foundation for future research.

**Spoken Language Modeling.** Generative text pretraining has inspired a new family of speech generation models. By proposing to quantize self-supervised representations, Lakhotia et al. (2021) rephrased speech generation as a language modeling task. The discrete tokens function as phonetic units, due to their accessible phonetic information (Nguyen et al., 2022; Sicherman & Adi, 2023; Yeh & Tang, 2024), and serve as inputs to train a Transformer decoder. Borsos et al. (2023) combined these units with audio codec tokens (Zeghidour et al., 2022; Défossez et al., 2023) to capture finer acoustic details. Non-phonetic information has also been incorporated with phonetic units to capture style or prosody (Kharitonov et al., 2022; Nguyen et al., 2025). Chen et al. (2025); Zhang et al. (2024) and Défossez et al. (2024) only use units from audio codecs. Notably, both SpeechTokenizer and Moshi distill HuBERT and WavLM representations into their first quantizer to guide it toward capturing linguistic information; without this, they would only encode local acoustic content. Moshi even uses a split quantizer to disentangle semantic and acoustic tokens inside the codec. Despite their ability to learn linguistic structures (Dunbar et al., 2021), purely speech-based models have exhibited limited factual knowledge and reasoning abilities. This prompted the development of hybrid speech-text models (Hassid et al., 2023; Nguyen et al., 2025; Défossez et al., 2024; Cuervo et al., 2025; Maimon et al., 2025b). A parallel research direction focuses on improving the speech units themselves (Algayres et al., 2023; Baade et al., 2025), as speech representations with more accessible phonetic information significantly improves linguistic knowledge (Poli et al., 2024). In our work, we deliberately focus on pure spoken language modeling from raw audio to isolate and evaluate the specific contributions of our speech encoder.

## 3 Method

As illustrated in figure 1, SpidR leverages self-distillation and online clustering, making predictions at multiple layers of the network. It is based on DinoSR, but with a novel learning objective. The student layers directly predict the assignment given by the corresponding codebook, instead of having multiple prediction heads at the end of the student encoder, which avoids codebook collapse.

We first extract feature frames $\mathbf{x} = (\boldsymbol{x}_1, ..., \boldsymbol{x}_n)$ from a speech utterance, with $\boldsymbol{x}_i \in \mathbb{R}^d$, using a convolutional block. We sample a random mask $M \subset \{1, ..., n\}$, with the sampling procedure from Baevski et al. (2020), and build $\tilde{\mathbf{x}}$, a corrupted version of $\mathbf{x}$ where for each $i \in M$, $\boldsymbol{x}_i$ has been replaced by a learned mask embedding. The student encoder is a Transformer (Vaswani et al., 2017) with $L$ layers, trained to predict the pseudo-labels derived from a teacher at the masked positions. Let $\tilde{\mathbf{z}}^k = (\boldsymbol{z}_1^k, ..., \boldsymbol{z}_n^k)$ be the output of the student encoder at layer $k$ from $\tilde{\mathbf{x}}$. As in previous works, this is the output of the feed-forward network in the Transformer block, before the final residual connection and layer normalization. The prediction is done at the last $K$ layers of the encoder. The label prediction at frame $i$ and intermediate layer $k$ is

$$\tilde{\boldsymbol{y}}_i^k = \phi^k(\tilde{\boldsymbol{z}}_i^k) \in (0, 1)^V, \tag{1}$$

where $\phi^k$ is the prediction head at layer $k$, with $L - K \le k \le L$, and $V$ is the number of labels. The prediction head is made of a single linear projection followed by a softmax. To derive the pseudo-labels, we first feed the unmasked frames $\mathbf{x}$ to the teacher. Let $\mathbf{z}^k$ be the output of the teacher encoder at intermediate layer $k$ after instance normalization.

Table 1: Summary of evaluation metrics. In section 4.3, we evaluate speech representations with ABX discriminability over triphones and MAP over words. We then compute in section 4.4 the quality of the derived discrete units and the downstream SLM performance at the lexical, syntactic, and semantic levels.

| Metric | Level | Task | Example / Interpretation |
|---|---|---|---|
| **Continuous representations** | | | |
| ABX | Phonetic | $d(x,a) < d(x,b)?$ $a \in A, b \in B,$ $x \neq a \in A.$ | Within speaker: $(\text{bat}_{s_1}, \text{bet}_{s_1}, \text{bat}_{s_1})$ Across speaker: $(\text{bat}_{s_1}, \text{bet}_{s_1}, \text{bat}_{s_2})$ |
| MAP words | Lexical | $\frac{1}{R}\sum_{1\leq i \leq R} P(i)$ | Average precision of retrieving same-word embeddings. |
| **Discrete units quality** | | | |
| ABX | Phonetic | $d(x,a) < d(x,b)?$ $a \in A, b \in B,$ $x \neq a \in A.$ | Same as ABX on continuous representations but with one-hots. |
| PNMI | Phonetic | $I(y;z)/H(y)$ | How much knowing the units $z$ reduces uncertainty about the gold phonemes $y$. |
| **Spoken language modeling** | | | |
| sWUGGY | Lexical | $p(a) > p(b)?$ | (brick, blick) |
| sBLIMP | Syntactic | $p(a) > p(b)?$ | (dogs eat meat, dogs eats meat) |
| tSC | Semantic | $p(a) > p(b)?$ | (paragraph with coherent ending, same but with uncoherent ending) |

The one-hot target label at frame $i$ and layer $k$ is

$$\boldsymbol{y}_i^k \in \{0,1\}^V \text{ where for } 1 \leq v \leq V, (\boldsymbol{y}_i^k)_v = \begin{cases} 1 & \text{if } v = \arg\min_{1 \leq u \leq V} \|\boldsymbol{z}_i^k - \mathbf{C}_u^k\|_2 \\ 0 & \text{otherwise} \end{cases}, \tag{2}$$

where $\mathbf{C}^k$ is the codebook associated to layer $k$, with $V$ codewords. The model is trained to predict the target labels from the teacher on the masked positions by minimizing the cross-entropy

$$-\frac{1}{|M| \cdot K} \sum_{\substack{i \in M \\ L-K \leq k \leq L}} \boldsymbol{y}_i^k \log \tilde{\boldsymbol{y}}_i^k, \tag{3}$$

The teacher is updated with an exponential moving average (EMA) of the student: the update at step $t$ is $\theta_{\text{teacher}} \leftarrow \beta_t \theta_{\text{teacher}} + (1 - \beta_t)\theta_{\text{student}}$. Following Liu et al. (2023a) and Baevski et al. (2022), the positional embeddings of the teacher are copied from the student, not updated by EMA. All activated codewords are updated with an EMA of the teacher output embeddings:

$$\begin{aligned} \boldsymbol{s}_v^k &\leftarrow \begin{cases} \tau \boldsymbol{s}_v^k + (1-\tau)\sum_{i:(\boldsymbol{y}_i^k)_v=1} \boldsymbol{z}_i^k & \text{if } \{i \mid (\boldsymbol{y}_i^k)_v = 1\} \neq \emptyset, \\ \boldsymbol{s}_v^k & \text{otherwise}, \end{cases} \\ n_v^k &\leftarrow \begin{cases} \tau n_v^k + (1-\tau)\sum_{i:(\boldsymbol{y}_i^k)_v=1} 1 & \text{if } \{i \mid (\boldsymbol{y}_i^k)_v = 1\} \neq \emptyset, \\ n_v^k & \text{otherwise}, \end{cases} \\ \mathbf{C}_v^k &\leftarrow \frac{\boldsymbol{s}_v^k}{n_v^k}, \end{aligned} \tag{4}$$

where $\boldsymbol{s}_v^k$ is initialized randomly and $n_v^k$ to 1, and $\tau$ is a constant decay parameter.

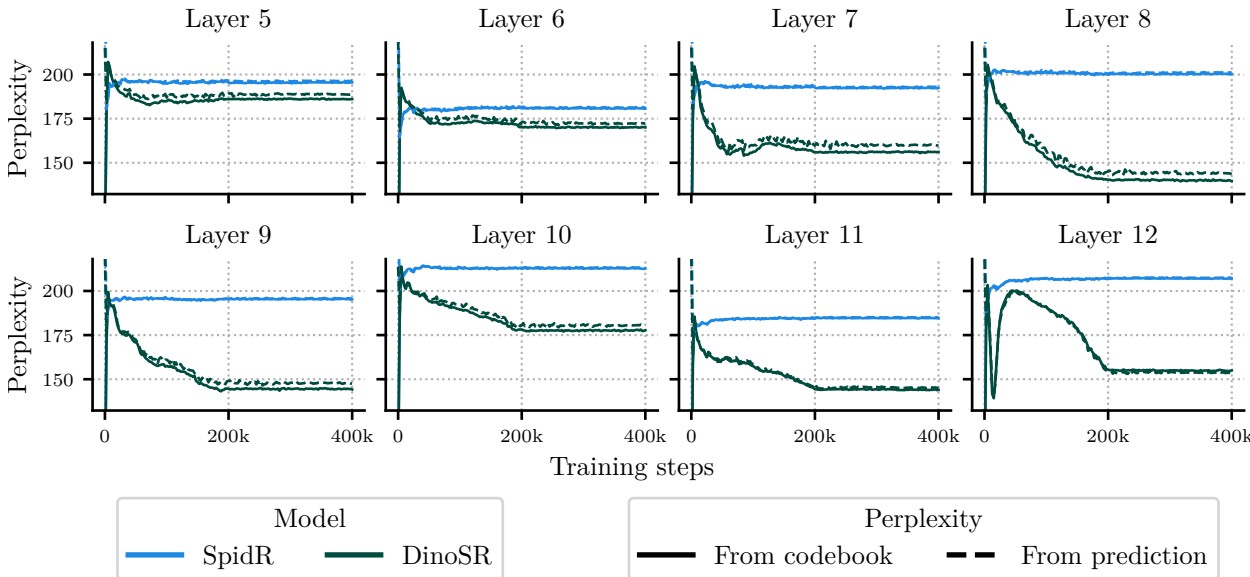

Figure 2: Codebook and prediction perplexities during training for SpidR and DinoSR on LibriSpeech `dev-clean`, with $K = 8$ codebooks. For each layer $k$, the codebook perplexity is computed over each batch with $\boldsymbol{p} = \boldsymbol{y}^k$ and then averaged across the dataset. The prediction perplexity uses $\boldsymbol{p} = \tilde{\boldsymbol{y}}^k$.

Note that with this update procedure, all embeddings $\boldsymbol{z}_i^k$ are used to update the codewords, but the non-activated codewords do not move. The main change from Liu et al. (2023a) is that the predictions are now aligned with the target layer. The output of layer $k$ of the student is used to predict the label derived from layer $k$ of the teacher, whereas DinoSR uses only the output of the last layer of the student encoder with $\tilde{\boldsymbol{y}}_i^k = \phi^k(\tilde{\boldsymbol{z}}_i^L)$. Baevski et al. (2022; 2023) also used the intermediate representations to train a SSL speech model, but in their case there was only one prediction head at the top of the student, trained to predict the average of the representations of the last $K$ layers of the teacher. Our approach is reminiscent of the deep supervision literature (Lee et al., 2015), but in a self-supervised learning context.

## 4  Experiments

We pretrain SpidR, compare its training stability to DinoSR in section 4.2, and evaluate the phonetic and word discriminability of its representations in section 4.3. We then extract discrete tokens and train spoken language models. In section 4.4, we show the improvement of SpidR on zero-shot spoken language modeling task over other SSL encoders in identical conditions. Finally, in section 4.5 we compare the training time of SpidR to that of HuBERT, DinoSR, and previous work on efficient SSL.

### 4.1  Setup

**Pretraining.**  The architecture follows the standard backbone from Baevski et al. (2020), and we make minimal changes from DinoSR. The model has a feature extractor with seven temporal convolutions and a projection layer, downsampling the 16 kHz input speech to 50Hz features of dimension $d = 768$. The student and teacher are BASE size Transformer encoders with $L = 12$ layers. The prediction is done at the top $K = 8$ layers, using codebooks with $V = 256$ codewords. We pretrain with 960 hours of speech from LibriSpeech (Panayotov et al., 2015). To maintain a fair comparison with DinoSR, we keep the same total batch size of 63 minutes of audio across 16 GPUs. The codebook decay parameter is kept constant: $\tau = 0.9$. The student encoder and feature extractor are optimized with AdamW[1] (Loshchilov & Hutter, 2019) for 400k steps. We use the same learning rate scheduler as Liu et al. (2023a), with a warmup from $5 \times 10^{-6}$ to $5 \times 10^{-4}$ within

---

[1]Previous work in SSL for speech (Baevski et al., 2020; Hsu et al., 2021; Liu et al., 2023a) reported using Adam, but the Adam optimizer in fairseq that was used is actually implemented as AdamW.

Table 2: Zero-shot evaluation of self-supervised speech representations (in %, chance level 50% for ABX). All models are trained on LibriSpeech 960h. For each model, the selected layer is the one with the lowest average ABX. The best scores are in **bold** and second best are underlined.

| Model | Layer | ABX within speaker ↓ | | ABX across speaker ↓ | | MAP words ↑ | |
|---|---|---|---|---|---|---|---|
| | | dev-clean | dev-other | dev-clean | dev-other | dev-clean | dev-other |
| wav2vec 2.0 | 6 | 4.47 | 5.63 | 5.25 | 7.82 | 44.81 | 31.92 |
| HuBERT | 11 | 3.38 | 4.26 | 4.01 | 6.49 | 46.07 | 33.37 |
| data2vec | 4 | 4.41 | 5.49 | 5.07 | 7.40 | 39.34 | 27.90 |
| data2vec 2.0 | 1 | 5.13 | 5.77 | 5.72 | 7.53 | **69.38** | 53.49 |
| DinoSR | 5 | 4.05 | 5.11 | 4.72 | 7.29 | 63.02 | 45.86 |
| DinoSR$^\dagger$ | 5 | 4.29 | 5.56 | 5.22 | 8.56 | 51.35 | 33.74 |
| SpidR | 6 | **3.32** | **3.74** | **3.66** | **4.95** | 66.50 | **55.26** |

$^\dagger$ Our re-implementation.

the first 12k steps, held constant until mid-training, and then exponentially decayed to $5 \times 10^{-6}$. We freeze the feature extractor after 200k steps. See appendix A in appendix for more details on the model and the hyperparameters.

We found during preliminary experiments that the norm of the weights of the $Q$, $K$, $V$ projections in the attention layers could increase along training, and potentially lead to spikes in the loss and model collapse. Removing the biases in those layers fixed this issue, with no negative impact. We also modify the schedule of the decay parameter of the teacher $\beta_t$. Instead of the warmup-and-constant schedule of Baevski et al. (2022) and Liu et al. (2023a), we take a smoother approach and set the decay at step $t$ to be $\beta_t = 1 - (1 - \beta_0) \exp(-t/T)$, where $T = 10000$ is a timescale parameter and $\beta_0 = 0.999$. See appendix C in appendix for an ablation from DinoSR to SpidR.

**Discrete units.** We extract the embeddings from the layer with the best phonetic discriminability. The output representations of this layer are then quantized to derive the discrete units. We consider two quantization methods. We first use vector quantization with K-means clustering (Nguyen et al., 2020; Lakhotia et al., 2021), training it with the `train-clean-100` subset of LibriSpeech. For DinoSR and SpidR, we also consider using the codebook predictions, by taking the assignment made by the prediction heads from the student encoder $\phi_k$ and selecting the label for which the probability is maximum. We deduplicate the tokens before passing them to the language model.

**Spoken language models.** The SLMs are OPT-125M models (Zhang et al., 2022), trained on the 6k hours subset of Libri-Light (Kahn et al., 2020) using fairseq2 (Balioglu et al., 2023). The architecture choice follows previous work (Hassid et al., 2023; Maimon et al., 2025a). We train on one A100 node with 8 GPUs, with a batch of at most 81920 tokens, and a context length of 2048 for 25k steps. The learning rate increases linearly to 1e−2 over 1000 steps, then follows a cosine annealing schedule. The other training parameters follow the defaults of OPT-125M. The selected checkpoint is the one with the lowest validation loss.

### 4.2 Preventing codebook collapse

Our motivation for changing DinoSR's learning objective was to stabilize the training procedure. We found in preliminary studies that the online clustering of DinoSR tended to collapse, as tracked by the codebook and prediction head perplexities. The perplexity $2^{H(\boldsymbol{p})}$, with $H(\boldsymbol{p}) = -\sum_{v \in V} \boldsymbol{p}_v \log_2 \boldsymbol{p}_v$ the entropy, measures the diversity of codewords used by the model, with $\boldsymbol{p}_v$ being the probability of the assignment $v$. The codebook perplexity at layer $k$ is measured with $\boldsymbol{p} = \boldsymbol{y}^k \in \{0, 1\}^V$, and the prediction head perplexity with $\boldsymbol{p} = \tilde{\boldsymbol{y}}^k \in (0, 1)^V$. With a perplexity of $V$, all codewords are used equally.

In figure 2, we compare the codebook and prediction perplexities of DinoSR and SpidR during training. The perplexities are computed on LibriSpeech `dev-clean` over each batch, using the same batch size as in

Table 3: Zero-shot discrete units quality and spoken language modeling metrics from wav2vec 2.0, HuBERT, WavLM Base, DinoSR, and SpidR (in %, chance level 50%, except for PNMI). The speech encoders are trained on LibriSpeech 960h and the language models on Libri-Light 6k. The vocabulary size is $V = 256$. For each model, the selected layer is the one with the lowest average ABX on continuous embeddings. The best scores are in **bold** and second best are underlined.

| Model | Layer | Units | Discrete units quality | | Language modeling | | | |
| | | | ABX ↓ | PNMI ↑ | sWUGGY ↑ | | sBLIMP ↑ | tSC ↑ |
| | | | | | all | in-vocab | | |
| wav2vec 2.0 | 6 | K-means | 9.33 | 0.609 | 62.29 | 68.50 | 53.34 | 65.97 |
| HuBERT | 11 | K-means | 7.32 | 0.637 | 65.50 | 73.67 | 55.60 | 68.75 |
| WavLM Base | 11 | K-means | 7.01 | **0.667** | 69.74 | 79.88 | 56.60 | 70.35 |
| DinoSR | 5 | Codebook | 7.92 | 0.620 | 60.10 | 64.56 | 57.04 | 69.44 |
| | | K-means | 10.87 | 0.558 | 56.69 | 59.42 | 54.32 | 65.76 |
| DinoSR[†] | 5 | Codebook | 7.73 | 0.614 | 58.93 | 62.07 | 55.79 | 66.08 |
| | | K-means | 10.54 | 0.591 | 55.73 | 57.58 | 54.04 | 63.68 |
| SpidR | 6 | Codebook | **6.31** | 0.602 | 69.78 | 79.98 | 58.10 | 70.14 |
| | | K-means | 7.20 | 0.636 | **71.89** | **82.46** | **59.48** | **70.46** |

[†] Our re-implementation.

pretraining, and then averaged across the dataset. Liu et al. (2023a) report that DinoSR has a much higher perplexity than other online clustering methods, such as VQ-APC (Chung et al., 2020) and Co-training APC (Yeh & Tang, 2022). However, DinoSR is still prone to codebook collapse, especially in the last layers. In DinoSR, the output of the last layer $\tilde{z}^L$ is given to all heads $\phi^k$ to derive the pseudo-labels from the intermediate layers of the teacher. The codebook assignments information for all $K$ layers must be linearly extractable from $\tilde{z}^L$. SpidR is more straightforward: $\tilde{z}^k$ is used to predict the assignments from layer $k$. This result suggests that our training objective reduces the distribution shift between the embeddings and the codebooks, a challenge frequently encountered in neural networks with vector quantization (Huh et al., 2023).

### 4.3 Evaluation of the learned speech representations

In order to train a spoken language model, we derive discrete units from the representations of the SSL model. For successful language modeling, the units need to encode the underlying linguistic content, not the speaker information or the acoustic background. Therefore, we want the model to have highly accessible phonetic and word information in its representations, and a well clustered representation space. Following previous work (Nguyen et al., 2020), we evaluate the SSL models with metrics computing the discriminability of the embeddings. This evaluation is then used to select the target layer for spoken language modeling (Lakhotia et al., 2021). We summarize the metrics used in this work in table 1.

The first metric of interest is the ABX discriminability over phonemes (Schatz, 2016). It measures how well triphones differing only by the central phone (like /bag/ and /beg/) are discriminated in the embedding space by comparing the distances between two instances $x$ and $a$ of the same triphone to the distance between $x$ and another triphone $b$. The test is successful if the representations of $x$ and $a$ are closer than those of $x$ and $b$. In the *within* speaker task, $a$, $b$ and $x$ are from the same speaker, whereas in the *across* speaker task, $a$ and $b$ are from the same speaker and $x$ from another one. We use the implementation of Poli et al. (2025) to compute ABX scores. It fixes issues with the slicing of features that existed in the Libri-Light version, which explains the differences with the scores reported by Liu et al. (2023a).

In addition to the ABX, which operates at the triphone level, we evaluate embedding discriminability at the word level. An ABX task where $A$ and $X$ are instances of the same word and $B$ is from a different word would be too easy in most cases. Instead, we opt for a more challenging metric: Mean Average Precision (MAP) over words (Carlin et al., 2011). This retrieval task requires that, for each word, the closest embeddings correspond to other instances of the same word. Unlike ABX, which uses Dynamic Time Warping to handle

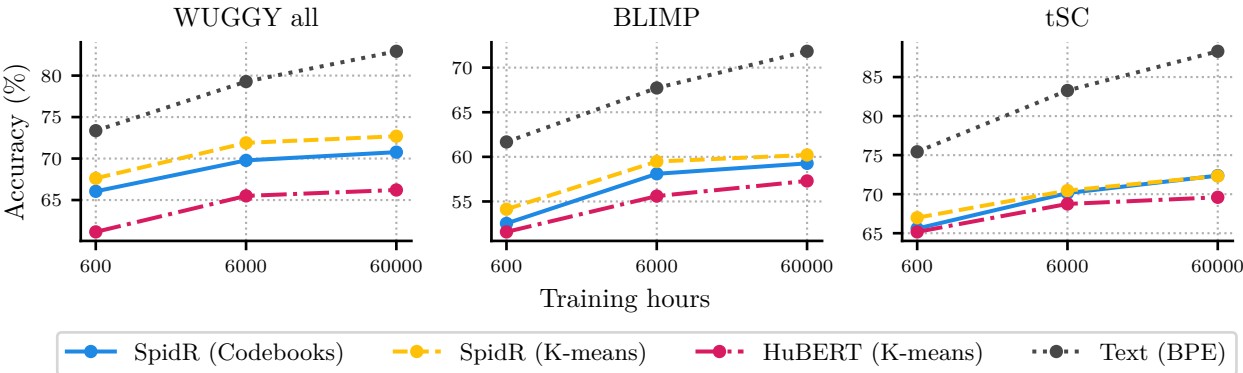

Figure 3: Data scaling results for a 125M parameters OPT model trained on Libri-Light, with different discrete units encoders. Zero-shot accuracy in %, chance level 50%. The speech encoders have $V = 256$ units. The log-likelihoods are normalized by the number of tokens, except for WUGGY with text.

duration differences between speech segments, we average word representations over the time axis. Following Algayres et al. (2020), we use MAP@$R$ (Musgrave et al., 2020) and get the final score by averaging over all words, where $R$ is the number of other instances of a given query word, and

$$\text{MAP@}R = \frac{1}{R}\sum_{i=1}^{R} P(i), \text{ where } P(i) = \begin{cases} \text{precision at } i & \text{if the } i\text{-th retrieval is correct,} \\ 0 & \text{otherwise.} \end{cases} \tag{5}$$

The intermediate layer chosen for each model is the one with the lowest average ABX error rate, which is not necessarily the best layer in terms of MAP (see figure 7 in appendix). As shown in table 2, SpidR outperforms baseline SSL models on both metrics. For all models, we computed the ABX using the angular distance on the representations of the intermediate layers, contrary to Liu et al. (2023a) who used the prediction heads with the KL-symmetric distance for DinoSR. See appendix B.1 in appendix for additional discriminability results and appendix B.2 for a visualization of the learned embeddings.

## 4.4 Evaluation of downstream spoken language modeling

**Evaluation metrics.** In order to assess the role of the speech encoder in spoken language modeling, we consider three standard tasks. At the lexical level, sWUGGY (Nguyen et al., 2020) evaluates the ability of the network to assign a higher probability to the true word than to a matching non-word. We also report results for "in-vocab" pairs, keeping only the words present in LibriSpeech. At the syntactic level, in sBLIMP, the network has to decide which sentence is grammatically correct, given minimal sentence pairs. Spoken StoryCloze (Mostafazadeh et al., 2017; Hassid et al., 2023) measures the ability of the model to choose the correct continuation of the beginning of a short story. We report the results for the "Topic" version (tSC), based on simpler negative examples. Following previous works, the log-likelihoods are normalized by the number of tokens. These metrics provide zero-shot evaluations of the model's linguistic knowledge; however they do not capture aspects related to non-linguistic or paralinguistic information (see de Seyssel et al. (2023); Maimon et al. (2025c) for complementary metrics). Note that we focus on the linguistic capabilities of the model, and our evaluation framework only uses log-likehods of sequences of discrete units. We do not train any vocoder in this work. Previous studies have shown that GANs can be trained to synthesize speech from discrete units of self-supervised speech encoders (Polyak et al., 2021), and can be conditioned on speaker, pitch, or style tokens (Nguyen et al., 2023; 2025).

**Comparison against other speech encoders.** To evaluate the contribution of units from SpidR for spoken language modeling, we compare in table 3 SLMs trained with units from wav2vec 2.0, HuBERT, DinoSR or SpidR on those three metrics. We also include WavLM BASE, even though it uses an additional robustness loss, and belongs to a slightly different class of SSL models. We keep a vocabulary size of $V = 256$

Table 4: Zero-shot spoken language modeling results (in %, chance level 50%) for ~150M parameters models trained on Libri-Light 6k from HuBERT or SpidR discrete units, across different number of units. Results for models based on HuBERT are from Chang et al. (2025); Messica & Adi (2024). The best scores are in **bold** and second best are underlined.

| Num. units | Model | Units | sWUGGY ↑ all | sWUGGY ↑ in-vocab | sBLIMP ↑ | tSC ↑ |
|---|---|---|---|---|---|---|
| 50 | HuBERT | K-means | - | 67.48 | 52.42 | 66.27 |
| | HuBERT | $\text{Spin}_{50}$ | 58.90 | 63.52 | 59.38 | 65.85 |
| | HuBERT | $\text{DC-Spin}_{50,4096}$ | 65.05 | 73.51 | **60.15** | 69.91 |
| | SpidR | K-means | **68.51** | **77.63** | 61.20 | **73.13** |
| 100 | HuBERT | K-means | - | 67.75 | 51.96 | 67.18 |
| | HuBERT | $\text{Spin}_{100}$ | 65.28 | 73.25 | 59.97 | 68.25 |
| | HuBERT | $\text{DC-Spin}_{100,4096}$ | 68.04 | 78.47 | **61.35** | 70.18 |
| | SpidR | K-means | **71.27** | **82.32** | 58.70 | **70.78** |
| 200 | HuBERT | K-means | - | 71.88 | 52.43 | 67.55 |
| | HuBERT | $\text{Spin}_{200}$ | 68.95 | 78.19 | **62.55** | 69.64 |
| | HuBERT | $\text{DC-Spin}_{200,4096}$ | 70.79 | 80.59 | 62.13 | 69.21 |
| | SpidR | K-means | **70.98** | **81.65** | 58.72 | **70.35** |
| 500 | HuBERT | K-means | 66.74 | 74.72 | 55.54 | 63.23 |
| | HuBERT | $\text{Spin}_{500}$ | 70.03 | 79.31 | 60.08 | 67.45 |
| | HuBERT | $\text{DC-Spin}_{500,4096}$ | **71.48** | 81.38 | **60.84** | 67.50 |
| | SpidR | K-means | 70.94 | **81.50** | 57.08 | **70.03** |

for all models to allow for exact comparison between units derived from K-means and from the codebook predictions. We also add an analysis of the discrete units' quality with the ABX on one-hot tokens, as well as the Phone Normalized Mutual Information (PNMI) (Hsu et al., 2021). The alignments used for PNMI are those from the ZeroSpeech 2021 challenge (Nguyen et al., 2020). Both metrics indicate how well the units correlate with the underlying phonemes. See appendix B.3 in appendix for a layer-wise analysis of the discrete units quality and downstream spoken language modeling results.SpidR outperforms all standard encoders on SLM metrics, and even outperforms WavLM Base when using units from K-means.

**Data scaling analysis.** To assess how the advantage of SpidR over other SSL models generalizes across different training conditions, we compare the scaling properties of SLMs trained with HuBERT or SpidR across varying data quantities in figure 3. We train SLMs on three dataset sizes: the 600h subset of Libri-Light, the 6k subset, or the full 60k dataset. We maintain the same hyperparameters as before, and we train for 150k steps when using the Libri-Light dataset instead of 25k steps, and for 15k steps on Libri-Light 600h. Additionally, we train a topline text LM using BPE tokens from the original books read, ensuring exact dataset matching between text and spoken LMs. The transcriptions are from Kang et al. (2024); on which we train a BPE tokenizer with 4096 tokens. Apart from the vocabulary size, all training hyperparameters match those of the SLMs. We evaluate the text LM on the original text versions of WUGGY, BLIMP and tSC. On WUGGY, we do not normalize log-likelihoods for the text LM since non-words are segmented into more tokens by the tokenizer. See appendix B.4 for DinoSR scaling properties.

Across all conditions, SpidR consistently outperforms HuBERT on all metrics, whether using codebook predictions or K-means clustering. However, it does not change the scaling properties: the scaling slopes are similar but SpidR has a constant advantage over HuBERT. Furthermore, text LMs trained under the same conditions achieve both better performance and superior scaling, particularly on tSC. Better performance on larger datasets could potentially be achieved with larger models.

Table 5: Pretraining compute footprint of SpidR against other SSL models operating at 50Hz. We report the pretraining times and total effective batch sizes in the default settings given in the corresponding papers. k2SSL Zipformer is trained using labels from the first iteration of HuBERT, and Academic HuBERT with labels from E-branchformer (Kim et al., 2023).

| Model | GPUs | Steps | Batch size | Pretraining time | GPU hours |
|---|---|---|---|---|---|
| HuBERT (Hsu et al., 2021) | A100 ×32 | 650k | 47 min | 62 hr | 1984 |
| DinoSR (Liu et al., 2023a) | V100 ×16 | 400k | 63 min | 180 hr | 2880 |
| data2vec 2.0 (Baevski et al., 2023) | A100 ×16 | 50k | 17 min | 43 hr | 688 |
| MelHuBERT (Lin et al., 2023a) | RTX 3090 ×1 | 630k | 8 min | 300 hr | 300 |
| Academic HuBERT (Chen et al., 2023a) | A100 ×8 | 1760k | 47 min | 240 hr | 1920 |
| k2SSL Zipformer (Yang et al., 2025) | V100 ×8 | 225k | 80 min | 64 hr | 513 |
| SpidR and DinoSR (our reimplem.) | A100 ×16 | 400k | 63 min | 23 hr | 369 |

**Across number of units.** Finally, we investigate the role of the number of units in the spoken LM in table 4. We train SLMs on SpidR units derived from K-means with vocabulary sizes in $\{50, 100, 200, 500\}$ under the same conditions as above. We compare the zero-shot scores to HuBERT-based models from Chang et al. (2025); Messica & Adi (2024). Those works use `transformer_lm_big` from fairseq (Ott et al., 2019) with 150M parameters, whereas we use the OPT-125M architecture. All language models are trained on Libri-Light 6k. The advantage of SpidR over HuBERT remains consistent across different vocabulary sizes. We also compare against units derived from HuBERT with Spin or DC-Spin. These approaches aim to improve speaker invariance and speech tokenization by learning auxiliary codebooks using swapped prediction. SpidR with standard K-means clustering matches the performance of HuBERT with DC-Spin units, with the latter showing advantages on sBLIMP, while SpidR performs better on the other metrics.

## 4.5 Codebase and pretraining time

In addition to learning strong phonetic representations, SpidR was designed with practical considerations in mind: reducing computational costs and simplifying the training pipeline. We developed a minimal PyTorch codebase compatible with the latest PyTorch features, with model implementations based on HuBERT from torchaudio (Hwang et al., 2023).

We re-implemented DinoSR in this codebase, reducing training time from the reported 180 hours to just 70 hours on 16 V100 GPUs under identical settings to Liu et al. (2023a). We further optimized the codebase for full compatibility with `torch.compile` (Ansel et al., 2024) and minimized host-device synchronization points. Since `torch.compile` merges native PyTorch modules and functions into optimized kernels, this results in significant throughput improvements. As shown in table 5, SpidR can be pretrained in under a day on 16 A100 GPUs. With 32 A100 GPUs, training SpidR only takes 14 hours (maintaining the same total batch size), compared to 62 hours for HuBERT. The single-pass training of SpidR also eliminates the feature extraction and label computation steps required by HuBERT, removing common engineering challenges. Figure 4 shows pretraining times for SpidR across different hardware configurations (4, 8, and 16 A100 or H100 GPUs) with constant total batch size. Using `torch.compile` provides approximately a 20% speedup in pretraining time. We open-source both the final checkpoints and the codebase.

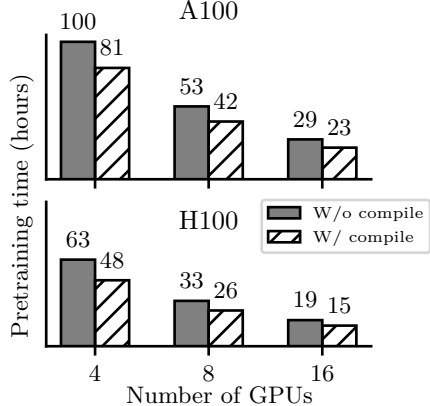

Figure 4: Approximate pretraining time for various hardware configurations with constant total batch size.

# 5    Conclusion

We presented SpidR: a self-supervised speech representation model that efficiently learns strong representations for spoken language modeling. We demonstrated that its learning objective, adapted from DinoSR, enables stable training and produces representations with salient phonetic information. Spoken language models using units derived from SpidR consistently outperform those based on HuBERT and DinoSR.

We developed SpidR by implementing minimal modifications to DinoSR to prevent codebook collapse. Future work may explore new architectures by iterating on the pairing between the student and the teacher, while preserving the core self-distillation and online clustering components. Our work focused on improving the linguistic capabilities of spoken language models through units that provide more accessible phonetic information. However, real-world speech systems require more than linguistic understanding. They must also preserve acoustic content and speaker information, capabilities that remain beyond the scope of our current approach. A possible next step would be to improve the encoder so that it learns not only linguistic units, but also disentangled representations for complementary aspects of the speech signal: prosodic (pitch, energy, duration), expressive (whispered, shouted, angry, sad, etc.), and speaker units. However, achieving disentanglement in a purely self-supervised approach remains a significant challenge (Polyak et al., 2021; Qian et al., 2022; Kharitonov et al., 2022; Lin et al., 2023b; Tu et al., 2024).

On a broader level, this work has implications for the design of spoken language models. Despite the emergence of generalist benchmarks such as SUPERB, modern self-supervised representation models have been designed with ASR as the primary objective, with every hyperparameter tuned accordingly. However, spoken language modeling has fundamentally different requirements, more closely aligned with the unsupervised term discovery tradition. Our findings demonstrate that textless SLM performs best when semantic information is readily accessible in the representations, rather than when ASR performance is maximized. Works that try to add speech modality to text-based LLMs often plug representations from the Whisper encoder into the LLM (Tang et al., 2024; Held et al., 2025; Xu et al., 2025) through an adapter. Our study suggests that this may be suboptimal, and that representations specifically designed for spoken language modeling deserve further exploration.

This work addressed exclusively English, and only with data from LibriVox audiobooks. Major multilingual SSL models are based on either wav2vec 2.0 (Conneau et al., 2021; Babu et al., 2022; Pratap et al., 2024) or HuBERT/WavLM (Chen et al., 2023b; 2024; Zanon Boito et al., 2024), and require massive computational resources for training. SpidR offers a solution for learning strong representations much faster, serving as foundation for future models and making approaches in other languages or multilingual settings more accessible due to reduced computational cost. Future work will focus on scaling the speech encoder to more data and languages while ensuring robustness to diverse acoustic conditions, with the goal of building a speech encoder capable of learning linguistic representations from ecological speech.

### Acknowledgments

This work was performed using HPC resources from GENCI-IDRIS (Grant 2023-AD011014368) and was supported in part by the Agence Nationale pour la Recherche (ANR-17-EURE-0017 Frontcog, ANR10-IDEX-0001-02 PSL*). M. P. acknowledges Ph.D. funding from Agence de l'Innovation de Défense. E.D. in his EHESS role was funded by an ERC grant (InfantSimulator). Views and opinions expressed are those of the authors only and do not necessarily reflect those of the European Union or the European Research Council. Neither the European Union nor the granting authority can be held responsible for them.

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

Table 6: SpidR pretraining hyperparameters. We trained with 16 A100 GPUs in our default setting.

| Parameter | Value |
|---|---|
| **Model** | |
| Conv1d dimension | 512 |
| Conv1d [(kernel size, stride)] | $[(10, 5)] + [(3, 2)] \times 4 + [(2, 2)] \times 2$ |
| Conv1d bias | False |
| Conv1d normalization | LayerNorm |
| Projection dropout | 0 |
| Positional encoding layers | 5 |
| Positional encoding total kernel size | 95 |
| Positional encoding groups | 16 |
| Hidden dimension $d$ | 768 |
| Number of Transformer layers $L$ | 12 |
| Number of attention heads | 12 |
| Transformer dropout | 0.1 |
| Attention dropout | 0.1 |
| Feed-forward dimension | 3072 |
| Feed-forward dropout | 0 |
| Layer drop probability | 5% |
| LayerNorm mode | After |
| $Q$, $K$, $V$ projection biases | False |
| Number of codebooks $K$ | 8 |
| Codebook decay $\tau$ | 0.9 |
| Codebook size $V$ | 256 |
| Initial decay of teacher $\beta_0$ | 0.999 |
| Decay timescale $T$ | 10 000 |
| Decay of teacher at step $t$ | $1 - (1 - \beta_0) \exp(-t/T)$ |

| Parameter | Value |
|---|---|
| **Optimizer** | |
| Name | AdamW |
| Peak learning rate | $5 \times 10^{-4}$ |
| Betas | $(0.9, 0.95)$ |
| Weight decay | 0.01 |
| Epsilon | $1 \times 10^{-6}$ |
| Max. gradient norm | 10 |
| Warmup steps | 12 000 |
| Hold steps | 188 000 |
| Decay steps | 200 000 |
| Conv. freeze step | 200 000 |
| **Data** | |
| Min. sequence length | 2000 |
| Max. sequence length | 320 000 |
| Max. samples in batch | 3 800 000 |
| Number of buckets | 1000 |
| Padding | False |
| Random crop | True |
| **Masking** | |
| Start probability | 8% |
| Span length | 10 |
| With overlap | True |

# A  Implementation details

## A.1  SpidR pretraining

Table 6 contains the full list of pretraining hyperparameters and figure 5 illustrates the two schedules that occur during training: the learning rate schedule and the EMA decay schedule of the teacher.

The positional encodings of DinoSR and SpidR are the same as those used by Baevski et al. (2022), and differ from Baevski et al. (2020); Hsu et al. (2021). Instead of only one convolutional layer with a large kernel size, they are made of 5 layers, each with a kernel size of $95/5 = 19$.

Batches are sampled using the following procedure. Audio files from LibriSpeech are first sorted and grouped into buckets by length, with only samples within the same bucket shuffled together. Batches are formed by selecting audio files from a given bucket until the target maximum number of samples in a batch is reached. If the target is not met, we continue filling the batch using files from the next bucket. No padding is applied, and audio samples longer than the maximum sequence length are randomly cropped.

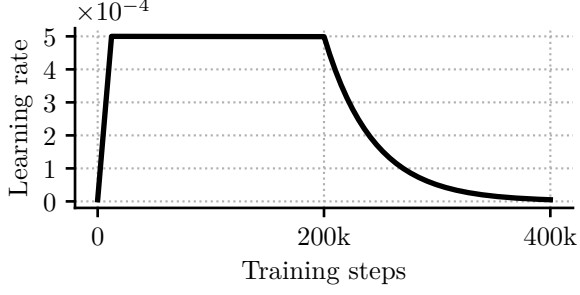 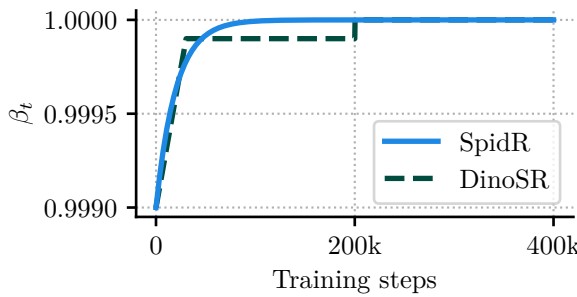

Figure 5: Learning rate schedule (left) and EMA decay schedule of the teacher for DinoSR and SpidR (right).

Table 7: ABX error rate of the codebook predictions (in %, chance level 50%, KL-symmetric distance). All models have codebooks of size 256. The best scores are in **bold** and second best are underlined.

| Model | Layer | ABX within speaker ↓ | | ABX across speaker ↓ | |
|---|---|---|---|---|---|
| | | dev-clean | dev-other | dev-clean | dev-other |
| DinoSR | 5 | 3.48 | 3.89 | 3.80 | 4.89 |
| DinoSR$^\dagger$ | 5 | 3.18 | 3.92 | 3.48 | 4.99 |
| HuBERT + Spin$_{256}$ | - | 3.85 | 5.20 | 4.32 | 6.36 |
| WavLM + Spin$_{256}$ | - | 4.41 | 4.67 | 4.80 | 5.82 |
| SpidR | 6 | **2.99** | **3.48** | **3.35** | **4.56** |

$^\dagger$ Our re-implementation.

## A.2 Masking procedure

We follow the masking procedure of Baevski et al. (2020), with parameters of Liu et al. (2023a), to sample the mask $M$, as shown in figure 6. We first extract features $\mathbf{x}$ of shape $(n, d)$ with $d = 768$ from the audio signal using the downsampling module. The masking process works as follows: each frame $i \in \{1, ..., n\}$ has an 8% probability of starting a mask span of length 10. Mask spans can overlap, and the proportion of masked frames depends on the total number of frames $n$.

Using the parameters from table 6, the average sequence length in a LibriSpeech 960h batch is 216000, corresponding to 13.5 seconds of audio and to $n = 675$ frames. For a typical 13.5-second audio sample, approximately 43% of all time-steps are masked, with an average span length of 11.9 frames, corresponding to 238ms of audio, a median of 8 frames, and a maximum of about 50 frames. For reference, the average triphone duration in LibriSpeech `dev-clean` and `dev-other` is 237ms, based on the annotations from Nguyen et al. (2020).

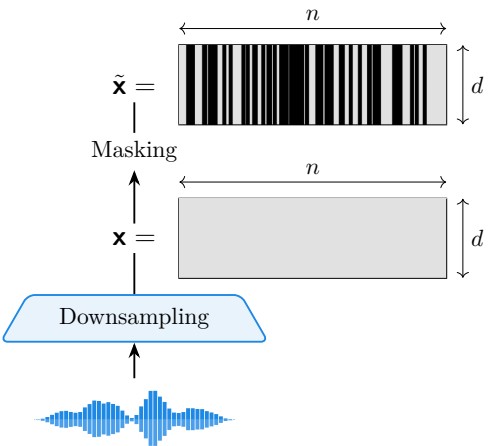

Figure 6: Masking procedure. The masked frames are in black, and the unmasked ones in gray.

# B Additional results

## B.1 Discriminability of continuous embeddings

In table 7, we compute ABX discriminability on the softmax outputs from either the prediction heads or the Spin codebooks. Instead of using the standard angular distance, we use the symmetrized KL divergence. This metric was used in Liu et al. (2023a) to evaluate DinoSR. We select the Spin checkpoints from Chang et al. (2023) with the same codebook size as DinoSR and SpidR.

We evaluate the phoneme and word discriminability of continuous embeddings from a wide range of monolingual English speech models in table 8. All BASE size models were trained on LibriSpeech and all LARGE models on Libri-Light. For each model, we select the best-performing layer in terms of average ABX score. We distinguish between standard self-supervised models (including SpidR), self-supervised models with additional robustness losses such as WavLM (Chen et al., 2022), and supervised models. We conducted a preliminary experiment where we fine-tuned SpidR with our own implementation of Spin.

Overall, masked prediction using discrete targets produces representations with salient phonetic information, and additional losses promoting invariance to acoustic and speaker conditions further improve performance. Supervision does not necessarily help—Whisper exhibits poor ABX scores, likely because it learns from multiple tasks simultaneously, making phonetic information less salient in its encoder representations.

Table 8: Evaluation of continuous representations from monolingual English speech models. All operate at a 50Hz framerate, except MR-HuBERT mono-base 25Hz and Conformer which are at 25Hz. Apart from Whisper, WavLM Base+ and WavLM Large, all Base size models were trained on LibriSpeech and all Large ones on Libri-Light. The best scores are in **bold** and second best are underlined.

| Model | Layer | ABX within speaker ↓ | | ABX across speaker ↓ | | MAP words ↑ | |
|---|---|---|---|---|---|---|---|
| | | dev-clean | dev-other | dev-clean | dev-other | dev-clean | dev-other |
| **Self-supervised models** | | | | | | | |
| wav2vec 2.0 (Baevski et al., 2020) | 6 | 4.47 | 5.63 | 5.25 | 7.82 | 44.81 | 31.92 |
| wav2vec 2.0 Large (Baevski et al., 2020) | 16 | 4.35 | 5.14 | 5.20 | 7.29 | 46.10 | 34.21 |
| HuBERT (Hsu et al., 2021) | 11 | 3.38 | 4.26 | 4.01 | 6.49 | 46.07 | 33.37 |
| HuBERT Large (Hsu et al., 2021) | 24 | 3.90 | 4.30 | 4.49 | 5.92 | 68.99 | 59.11 |
| HuBERT Extra Large (Hsu et al., 2021) | 48 | 4.04 | 4.38 | 4.76 | 6.21 | 64.64 | 54.64 |
| data2vec (Baevski et al., 2022) | 4 | 4.41 | 5.49 | 5.07 | 7.40 | 39.34 | 27.90 |
| data2vec Large (Baevski et al., 2022) | 7 | 4.51 | 5.46 | 5.20 | 7.15 | 38.56 | 28.70 |
| data2vec 2.0 (Baevski et al., 2023) | 1 | 5.13 | 5.77 | 5.72 | 7.53 | 69.38 | 53.49 |
| data2vec 2.0 Large (Baevski et al., 2023) | 2 | 6.68 | 6.55 | 7.41 | 8.43 | 66.85 | 55.42 |
| Eh-MAM (Seth et al., 2024) | 1 | 4.31 | 5.36 | 4.96 | 7.58 | 60.63 | 42.80 |
| MR-HuBERT mono-base 50Hz (Shi et al., 2024) | – | 3.64 | 3.99 | 4.21 | 5.39 | **70.72** | **63.08** |
| MR-HuBERT mono-base 25Hz (Shi et al., 2024) | – | 3.47 | 3.82 | 4.00 | 5.14 | 68.50 | 61.58 |
| DinoSR (Liu et al., 2023a) | 5 | 4.05 | 5.11 | 4.72 | 7.29 | 63.02 | 45.86 |
| DinoSR[†](Liu et al., 2023a) | 5 | 4.29 | 5.56 | 5.22 | 8.56 | 51.35 | 33.74 |
| SpidR | 6 | **3.32** | **3.74** | **3.66** | **4.95** | 66.50 | 55.26 |
| **With self-supervised robustness loss** | | | | | | | |
| WavLM Base (Chen et al., 2022) | 11 | 3.03 | 3.71 | 3.50 | 5.21 | 71.57 | 58.09 |
| WavLM Base+ (Chen et al., 2022) | 12 | 3.54 | 4.08 | 4.08 | 5.82 | 62.77 | 51.63 |
| WavLM Large (Chen et al., 2022) | 24 | 3.94 | 4.32 | 4.62 | 5.99 | 67.56 | 57.49 |
| ContentVec$_{100}$ (Qian et al., 2022) | 12 | 3.29 | 4.04 | 3.83 | 5.49 | 63.85 | 52.17 |
| HuBERT + Spin$_{2048}$ (Chang et al., 2023) | 12 | **2.70** | **3.23** | **3.05** | **4.08** | 68.70 | 61.41 |
| WavLM + Spin$_{2048}$ (Chang et al., 2023) | 12 | 3.05 | 3.51 | 3.52 | 4.44 | **75.20** | **67.13** |
| SpidR + Spin$_{2048}$ | 12 | 2.73 | 3.51 | 3.11 | 4.47 | 61.33 | 50.99 |
| **With supervision** | | | | | | | |
| Whisper small.en (Radford et al., 2023) | 8 | 7.03 | 8.21 | 8.27 | 11.81 | 16.35 | 11.03 |
| Conformer ASR[‡] (Gulati et al., 2020) | 8 | 2.79 | 3.76 | 3.21 | 5.22 | 63.03 | 45.74 |
| wav2vec 2.0 ASR (Baevski et al., 2020) | 6 | 3.94 | 4.90 | 4.50 | 6.52 | 56.98 | 43.52 |
| wav2vec 2.0 Large ASR (Baevski et al., 2020) | 10 | 3.52 | 4.58 | 4.09 | 6.20 | 54.04 | 40.06 |
| HuBERT Large ASR (Hsu et al., 2021) | 14 | 4.69 | 5.38 | 5.48 | 7.27 | 43.25 | 33.09 |
| HuBERT + phoneme classif. (Poli et al., 2024) | 12 | **0.82** | **1.54** | **0.97** | **2.35** | 68.48 | 57.04 |

[†] Our re-implementation.

[‡] Using `speechbrain/asr-conformer-transformerlm-librispeech` (Ravanelli et al., 2024).

We compare in figure 7 the ABX and MAP on continuous embeddings by layer for HuBERT, DinoSR (both the original checkpoint and our replication) and SpidR. The ABX scores are averaged across subsets and speaker conditions, and MAP across the two subsets.

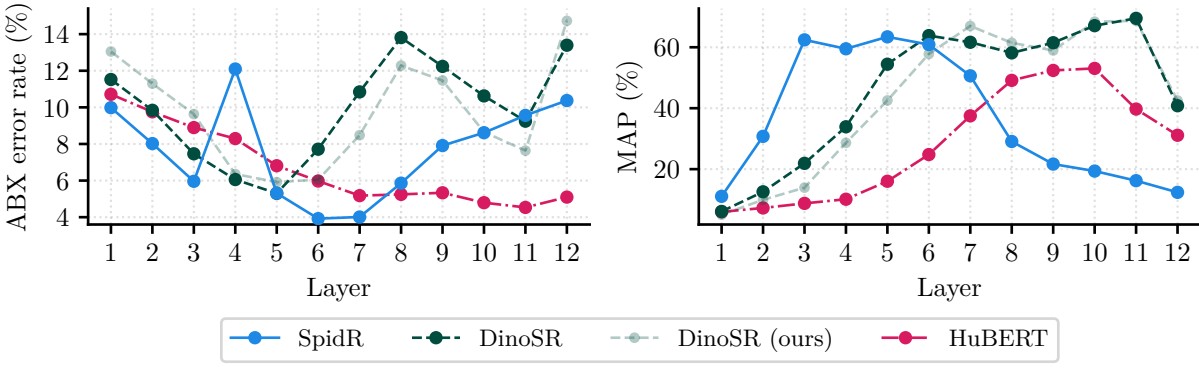

Figure 7: ABX and MAP (in %, chance level 50% for ABX) by layer for SpidR, DinoSR and HuBERT.

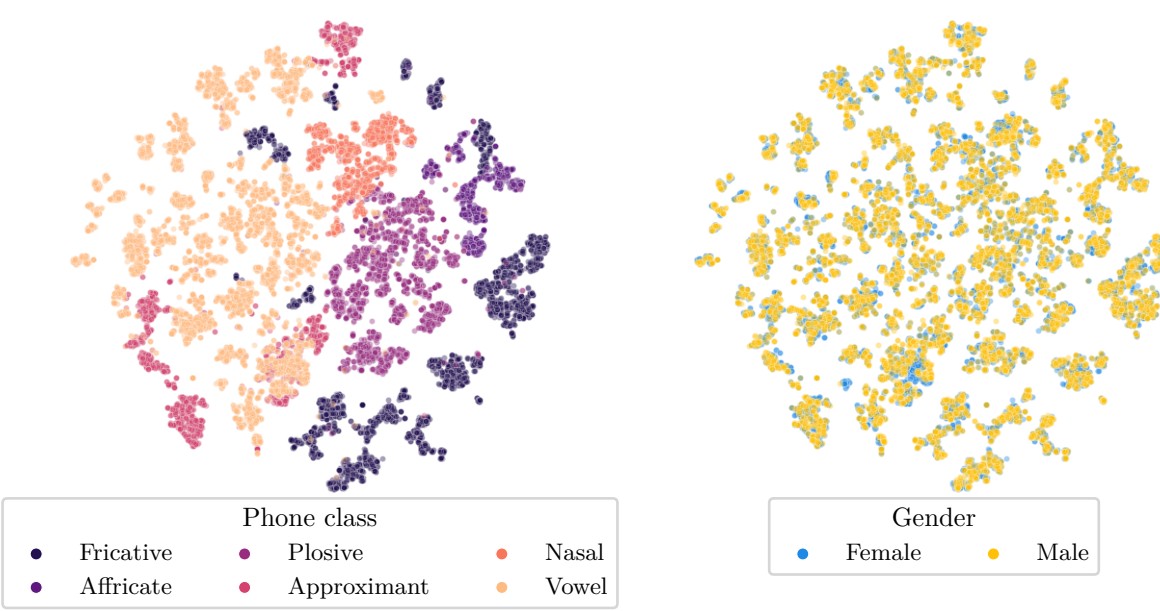

Figure 8: t-SNE visualization of phone embeddings from SpidR layer 6 on LibriSpeech `dev-clean`. Embeddings are colored by phone class (left) and by speaker gender (right).

## B.2 Embeddings visualization

We visualize the embedding space of SpidR in two dimensions using t-SNE (van der Maaten & Hinton, 2008), following de Seyssel et al. (2022). We train t-SNE on phone embeddings of LibriSpeech `dev-clean` from layer 6 of SpidR. For each speaker, we sample 10 instances per phone and average each embedding along the time dimension, resulting in approximately 15 000 samples.

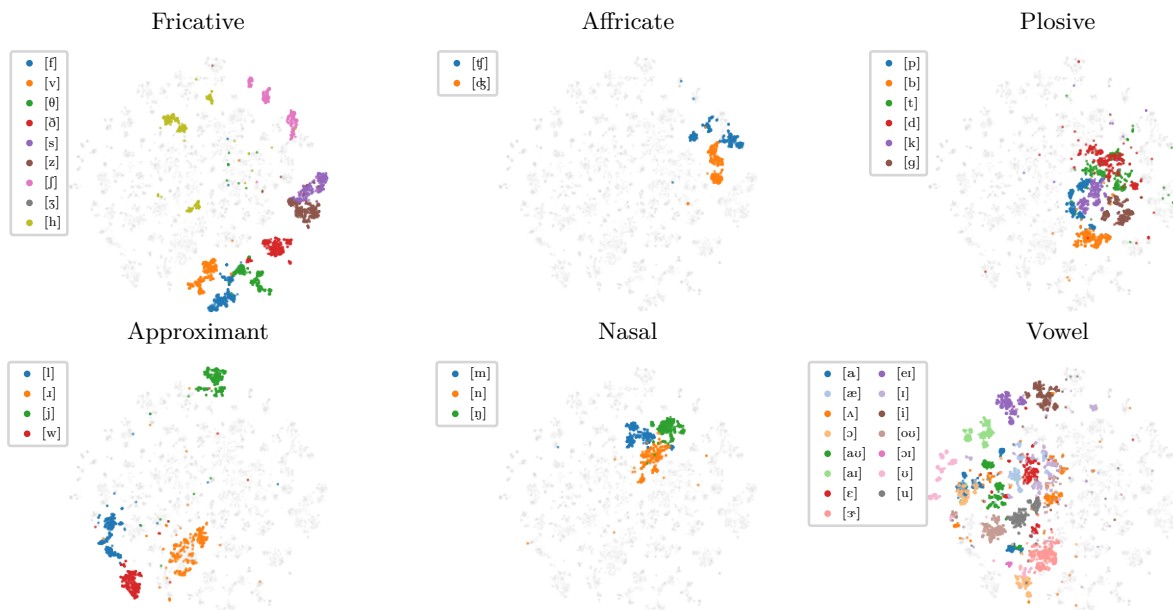

Figure 9: t-SNE visualization of phone embeddings from SpidR layer 6 on LibriSpeech `dev-clean`, colored by individual phones within each phone class. Embeddings from other classes are shown in gray.

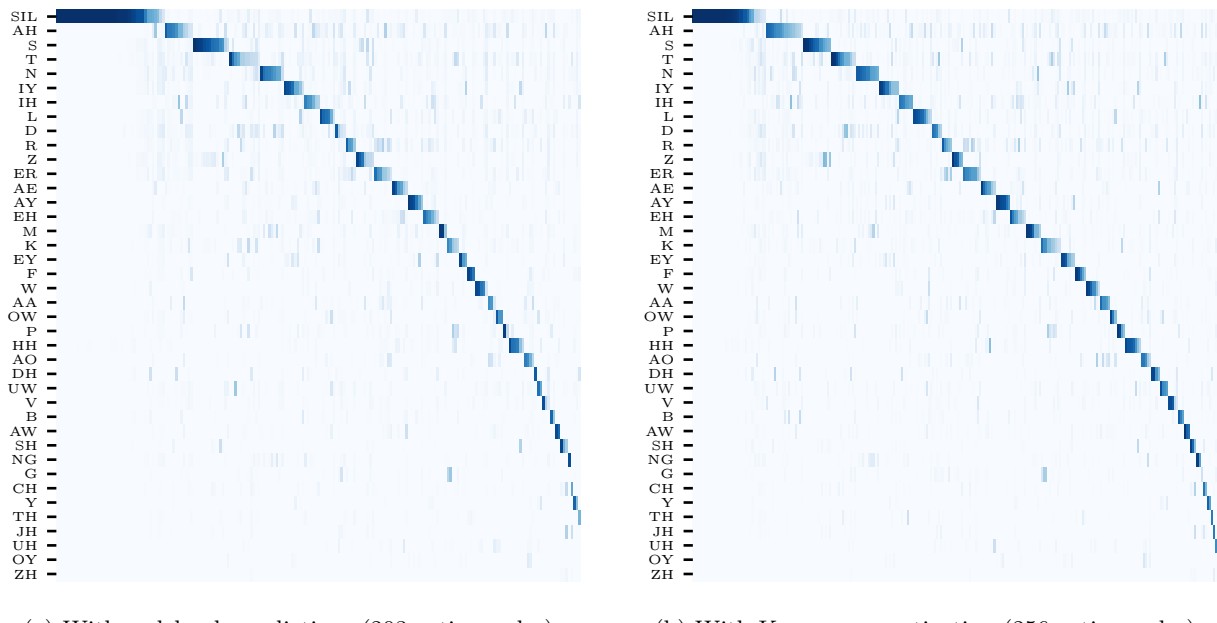

(a) With codebook predictions (203 active codes).

(b) With K-means quantization (256 active codes).

Figure 10: $\mathbb{P}(\text{phone} \mid \text{code})$ visualization for SpidR layer 6 using either codebook predictions (left) or K-means quantization (right), on LibriSpeech `dev-clean` and `dev-other`.

In figure 8, we color the embeddings by either the underlying phone class or by the speaker gender. For more fine-grained visualization, we color by individual phones within each phone class in figure 9. Overall, the embedding space is well clustered by phone class, and even by individual phone, whereas the speaker information is not directly extractable from the embeddings.

## B.3 Layer-wise analysis

In addition to the discrete units analysis in table 3, we compute in figure 11 the ABX discriminability and PNMI for other intermediate layers of SpidR and HuBERT, with units derived from codebook predictions or K-means quantization. As in section 4.1, the K-means are trained on LibriSpeech `train-clean-100`. Figure 10 shows the $\mathbb{P}(\text{phone} \mid \text{code})$, with codes from SpidR layer 6 on LibriSpeech `dev-clean` and `dev-other`. The vertical axes are sorted by phone frequency in the annotated data.

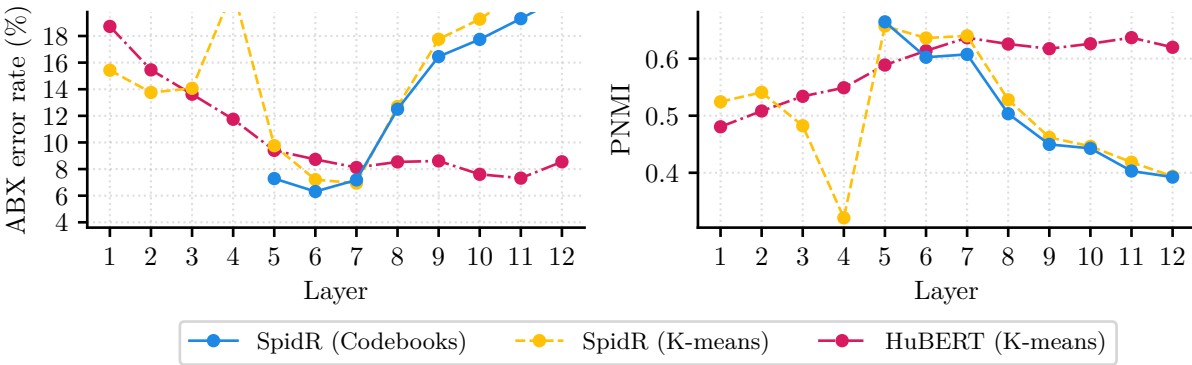

Figure 11: ABX (in %, chance level 50%) and PNMI by layer on discrete units from SpidR using codebook predictions or K-means, and from HuBERT using K-means, with $V = 256$ units. ABX scores averaged across subsets and speaker conditions, and PNMI computed on LibriSpeech `dev-clean` and `dev-other`.

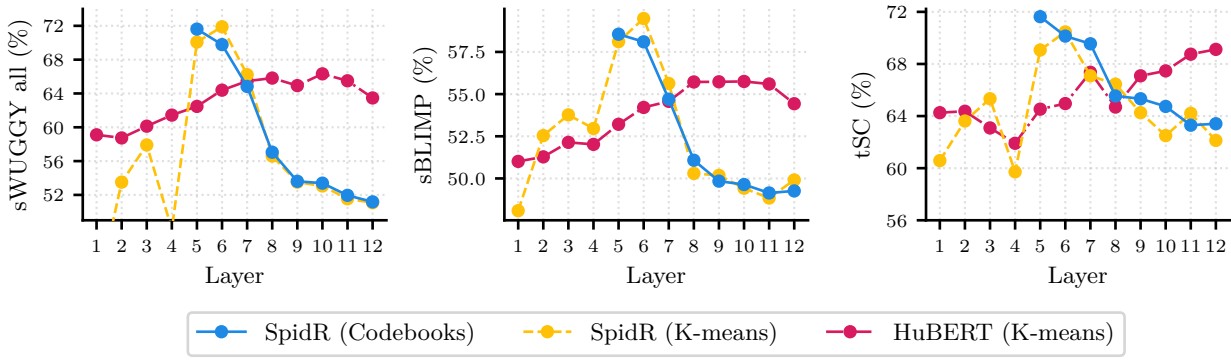

Figure 12: Zero-shot spoken language modeling from each layer of HuBERT and SpidR (in %, chance level 50%), with units from codebook predictions or from K-means quantization, with $V = 256$ units.

We also trained spoken language models from the units obtained from each intermediate layer in the same conditions as section 4.1. Figure 12 shows the accuracies on zero-shot spoken language modeling for the three encoders. Finally, to assess how well the zero-shot metrics serve as proxy tasks, we compare spoken language modeling scores against phonetic- and word-level metrics in figure 13 (continuous embeddings) and figure 14 (discrete units). We distinguish between SpidR using K-means units, where ABX is computed on standard embeddings, and SpidR using codebook predictions, where ABX is computed on codebook predictions with symmetric KL divergence. We compute Pearson correlation coefficients between each proxy metric and downstream evaluation score. Note that this analysis does not capture inter-model differences well, and that correlations are influenced by the fact that SpidR's final layers perform poorly across most metrics.

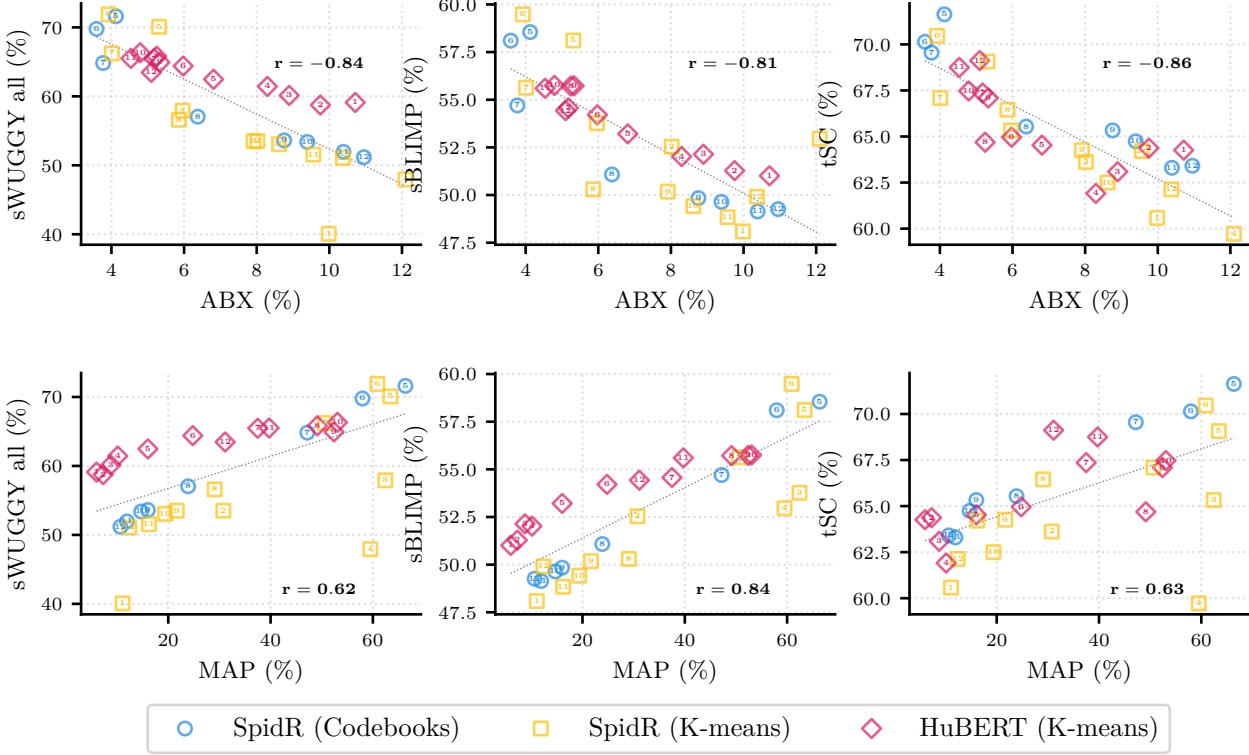

Figure 13: Spoken language modeling against discriminability of the continuous representations. Dots are labeled by intermediate layer index. ABX for SpidR (Codebooks) is computed over codebook predictions.

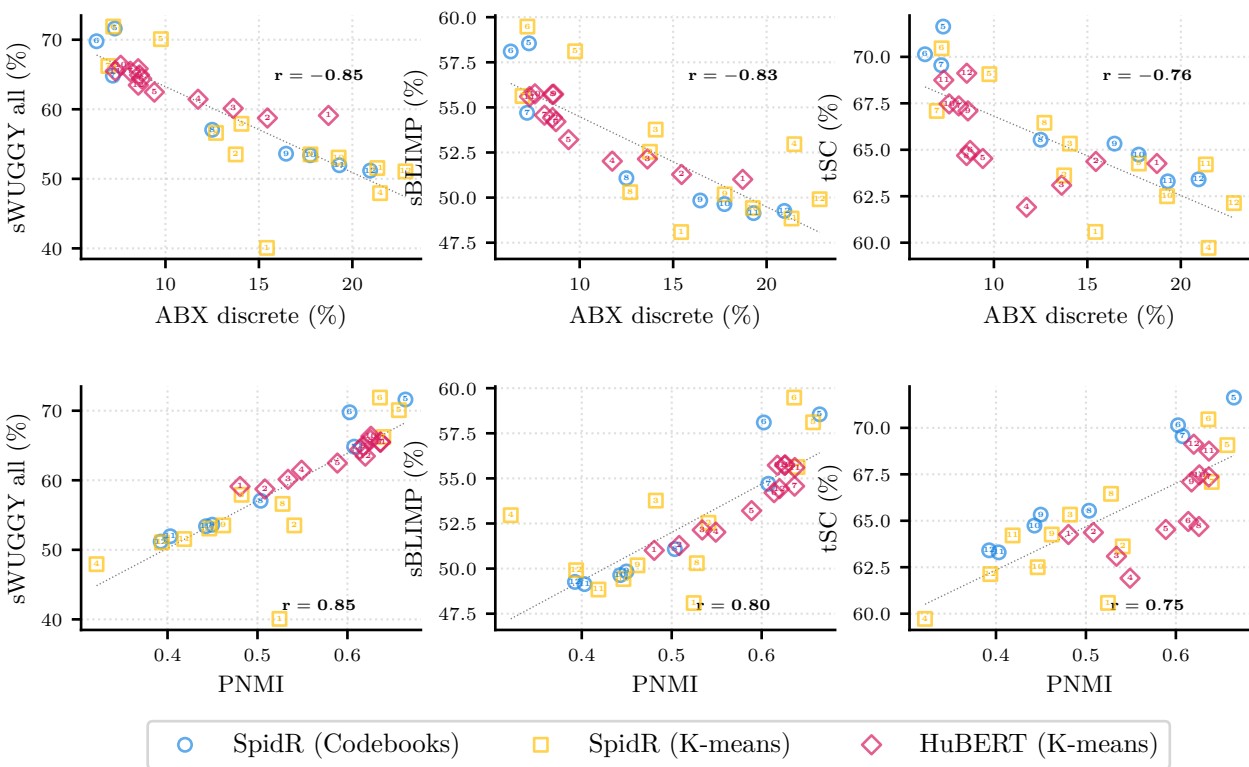

Figure 14: Spoken language modeling against phonetic evaluation of the discrete units, with $V = 256$ units. Dots are labeled by intermediate layer index.

## B.4 Data scaling analysis

We report in figure 15 the scaling properties of SLMs across varying data quantities, as in figure 3, with DinoSR in addition of SpidR and HuBERT.

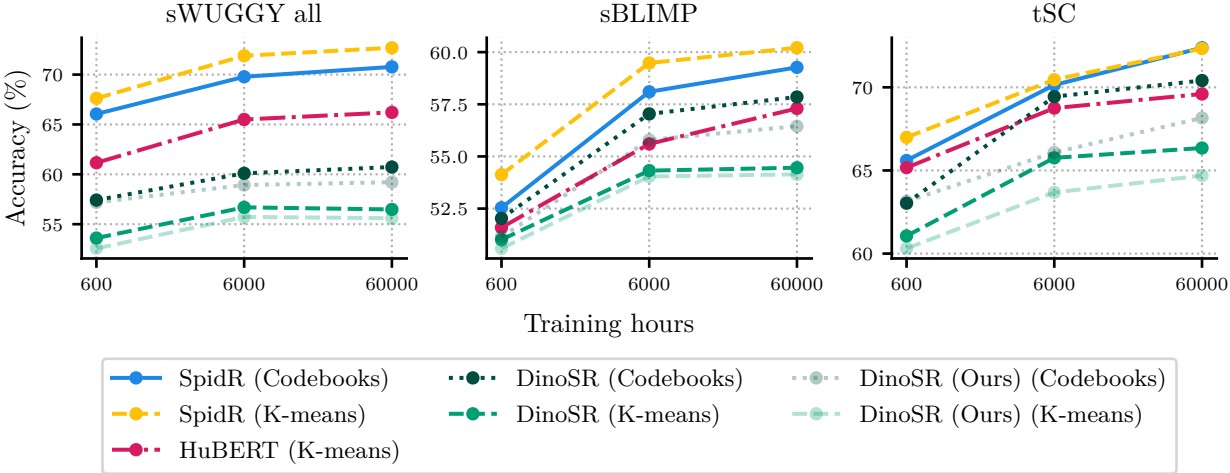

Figure 15: Data scaling results for a 125M parameters OPT model trained on Libri-Light, with different discrete units speech encoders. Zero-shot accuracy in %, chance level 50%. The speech encoders have $V = 256$ units. The log-likelihoods are normalized by the number of tokens.

## C   Ablation study

Table 9: Ablation study from DinoSR to SpidR (in % except for PNMI, chance level 50% for ABX). The discrete units are derived from the codebook predictions. The ABX scores are averaged across subsets and speaker conditions, MAP across the two subsets, and PNMI is computed on LibriSpeech `dev-clean` and `dev-other`.

| Model | Layer | Continuous embeddings | | Discrete units quality | |
|---|---|---|---|---|---|
| | | ABX ↓ | MAP ↑ | ABX ↓ | PNMI ↑ |
| DinoSR[†] | 5 | 5.91 | 42.55 | 7.73 | 0.614 |
| DinoSR + Heads | 7 | 4.22 | 63.39 | 7.87 | 0.610 |
| DinoSR + Exp. EMA | 6 | 5.86 | 51.85 | 8.77 | 0.609 |
| DinoSR + Heads + Exp. EMA = SpidR | 6 | 3.92 | 60.88 | 6.31 | 0.602 |

[†] Our re-implementation.

We developed SpidR by making two key changes from DinoSR. First, we modified the learning objective by adding prediction heads to the student's intermediate layers instead of using only the final layer. This showed promising results, but we noticed that the training loss would slightly increase mid-training, suggesting that the student was struggling to keep up with the teacher. To solve this problem, we modified the teacher's EMA decay schedule to follow a smoother trajectory that approaches 1 faster without plateauing at 0.9999.

In table 9 we ablate these two changes: "Heads" refers to the new learning objective, and "Exp. EMA" refers to the new shape of the EMA decay schedule. We evaluate both in terms of continuous embedding discriminability, and discrete units quality using the prediction heads.

## D   Downstream supervised evaluation

To provide a comprehensive evaluation of SpidR beyond our primary task of spoken language modeling, we also assess its performance on ASR benchmarks. We fine-tune both DinoSR and SpidR for ASR using the 1h and 10h labeled subsets of Libri-Light (Kahn et al., 2020), as well as the `train-clean-100` subset of LibriSpeech (Panayotov et al., 2015). We adopt the exact fine-tuning configuration from wav2vec 2.0 without additional hyperparameter tuning. We also run the phoneme recognition and ASR without LM tasks of SUPERB (wen Yang et al., 2021) with the default hyperparameters.

As shown in table 10, SpidR performs comparably to wav2vec 2.0 on clean evaluation sets but lags behind on the other two, while DinoSR surpasses all other models. This pattern persists in the ASR and phoneme recognition tasks from SUPERB, as shown in table 11. While both SpidR and WavLM have representations that discriminate phonemes the most (as shown in table 8), they are outperformed by data2vec or DinoSR on phoneme recognition with a linear probe over frozen features. Similarly, the advantage in terms of ABX discriminability does not translate into superior ASR performance, even after fine-tuning. This divergence between supervised classification and unsupervised clustering may due to the existence of subspaces dedicated to different types of information within the embeddings (Liu et al., 2023b). We illustrate the discrepancy between SLM and supervised evaluations of speech encoders in figure 16.

Table 10: Word Error Rate (in %) on LibriSpeech dev and test sets after finetuning on Libri-Light low-resource labeled splits. All models are pretrained on LibriSpeech 960h, and decoded greedily, without language model. The best scores are in **bold** and second best are underlined.

| Model | dev-clean | dev-other | test-clean | test-other |
|---|---|---|---|---|
| **1h labeled** | | | | |
| wav2vec 2.0 (Baevski et al., 2020) | 24.1 | 29.6 | 24.5 | 29.7 |
| HuBERT (Hsu et al., 2021)♠ | 20.2 | 28.1 | 20.6 | 28.9 |
| WavLM Base (Chen et al., 2022) | – | – | 24.5 | 29.2 |
| MR-HuBERT mono-base (Shi et al., 2024) | 18.8 | 23.7 | 19.3 | 24.5 |
| DinoSR (Liu et al., 2023a)* | **18.1** | **23.3** | **18.3** | **23.6** |
| DinoSR (Liu et al., 2023a)† | 18.4 | 23.7 | 18.6 | 23.7 |
| SpidR | 29.1 | 35.9 | 29.5 | 36.1 |
| **10h labeled** | | | | |
| wav2vec 2.0 (Baevski et al., 2020) | 10.9 | 17.4 | 11.1 | 17.6 |
| HuBERT (Hsu et al., 2021)♠ | 9.6 | 16.6 | 9.7 | 17.0 |
| WavLM Base (Chen et al., 2022) | – | – | 9.8 | 16.0 |
| MR-HuBERT mono-base (Shi et al., 2024) | 8.5 | 13.2 | 8.5 | 13.5 |
| DinoSR (Liu et al., 2023a)* | **7.1** | **12.1** | **7.2** | **12.5** |
| DinoSR (Liu et al., 2023a)† | 7.3 | 12.7 | 7.4 | 12.8 |
| SpidR | 10.7 | 19.4 | 11.0 | 19.8 |
| **100h labeled** | | | | |
| wav2vec 2.0 (Baevski et al., 2020) | 6.1 | 13.5 | 6.1 | 13.3 |
| HuBERT (Hsu et al., 2021)♠ | 5.8 | 12.9 | 5.8 | 12.8 |
| WavLM Base (Chen et al., 2022) | – | – | 5.7 | 12.0 |
| MR-HuBERT mono-base (Shi et al., 2024) | 4.9 | **9.0** | 4.9 | **9.2** |
| DinoSR (Liu et al., 2023a)* | **4.0** | 10.1 | **4.2** | 9.9 |
| DinoSR (Liu et al., 2023a)† | 4.2 | 10.3 | 4.4 | 10.2 |
| SpidR | 6.1 | 15.8 | 6.3 | 15.9 |

♠ From Shi et al. (2024).

* Original pretrained model that we fine-tuned.

† Our re-implementation (both pretraining and fine-tuning).

Table 11: Results of SSL models on SUPERB ASR and phoneme recognition tasks. All models are pretrained on LibriSpeech 960h. The best scores are in **bold** and second best are underlined.

| Model | Content | |
|---|---|---|
| | PR PER ↓ | ASR WER ↓ |
| wav2vec 2.0 (Baevski et al., 2020) | 5.74 | 6.43 |
| HuBERT (Hsu et al., 2021) | 5.41 | 6.42 |
| data2vec (Baevski et al., 2022) | 4.69 | 4.94 |
| data2vec 2.0 (Baevski et al., 2023) | 3.93 | 4.91 |
| WavLM Base (Chen et al., 2022) | 4.84 | 6.21 |
| MR-HuBERT mono-base (Shi et al., 2024) | 4.16 | 5.76 |
| DinoSR (Liu et al., 2023a) | **3.21** | **4.71** |
| DinoSR (Liu et al., 2023a)† | 3.54 | 5.01 |
| SpidR | 4.86 | 7.47 |

† Our re-implementation.

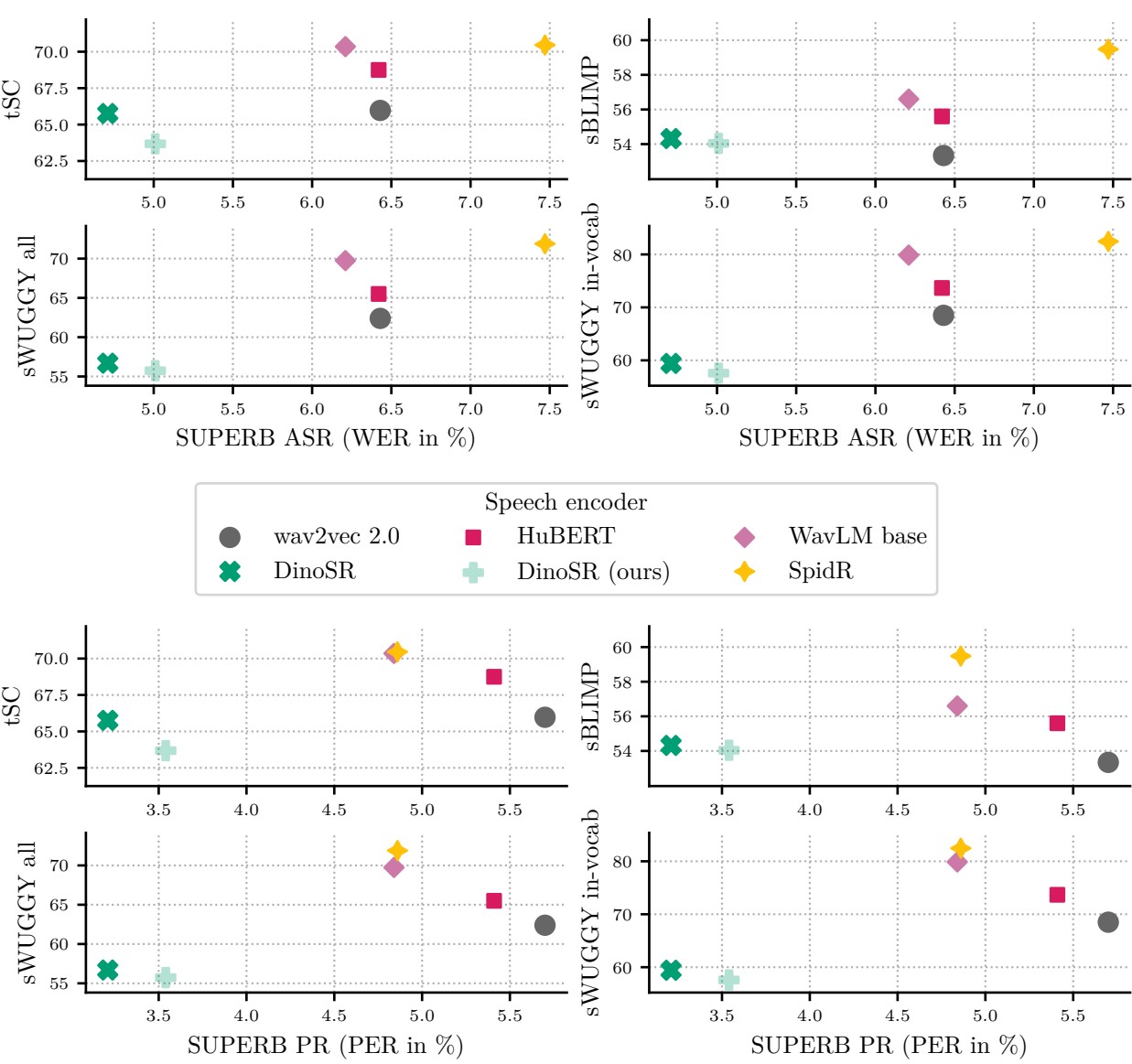

Figure 16: Spoken language modeling against supervised evaluation of speech encoders. The speech encoders are all trained on LibriSpeech, and the LMs are evaluated using $V = 256$ speech units from the best layer of each encoder in terms of ABX discriminability. The SUPERB evaluation is done via a linear probe learned on an average of the intermediate representations. This figure comprises the results from tables 3 and 10.

