# OpenReview forum: "SpidR: Learning Fast and Stable Linguistic Units for Spoken Language Models Without Supervision"
_TMLR — Accepted by TMLR_

### Review · Reviewer_3tfg · 2025-07-20

**Summary Of Contributions:**

This manuscript shows and will provide open source implementation of how in about 1 day of time and with access to 16 A100 GPUs you can build a competitive SSL model on 960 hours of publicly available read speech (LibriSpeech).

Specific contributions are:

1) layer-level teacher student training where each affected teacher layer provides targets to the corresponding student layer

2) code engineering

3) extensive evaluation

Regarding 1), this is not new (google shows https://arxiv.org/pdf/2210.01351 as an example) but appears not to be done with models explored in this work. Please add a bit info on what kind of layer-level training has been done prior to you.

Regarding 2), only limited details are provided but they sound quite reasonable.

Regarding 3), this is common in this line of work.

**Audience:**

Yes

**Claims And Evidence:**

Yes

**Requested Changes:**

1) It appears that you are following https://arxiv.org/pdf/2504.08528 to position your work as "speech language modelling". Without a proper introduction and explanation, I find this terminology/taxonomy of models confusing. Given how much nuances are there and variants to mix speech and language, could you please clearly define what they are and how they are different from models definitely known to the audience of these proceedings.

2) Could you please check your manuscript for grammar/semantics. I find phrasing used in many places to be borderline at best. For example,  "some approaches added a new step on the units" where I struggle to understand what this mean. Another example is "The selected checkpoint in the one with the lowest validation loss". There may be more than that.

3) I find discussion of terabytes (storage) and weeks (time) to be confusing and somewhat misleading as these units are directly linked with quantities of data used and amount of GPU resource available. Could you please present them with reference to some standard (per hour of speech, per 1xGPU)?

4) One and perhaps the most important claim in this manuscript is that it "makes pretraining significantly more stable and resistant to codebook collapse". I believe you present evidence for codebook collapse albeit it is averaged out in PPL across the data used to compute it. I do not believe I see any evidence presented of the previous methodologies displaying it. I also do not see how you measure stability and how it manifests itself in your experiments. I would also love to see some examples of its lack in existing methodologies. Simply saying that does not suffice.

5) Please make it very clear that your methodology requires a certain restriction on the nature of students/teachers (at least as many layers? do you require any other topological and other constraints?)

6) You have presented an evidence that pairing all layers of students and teachers is a good idea. Due to model-centric view (everything is a model), I am not fully certain if you have presented your model where only top layers are paired. I do not believe you have presented any evidence of using pairing for some layers only and it is not clear which layers are important to pair and which are not. Could you please explain why none (some) of that investigation has not been done?

7) I struggle to understand what "using the latest advancements of PyTorch" means.

8) I would be careful making statements such as "hackable foundation for future research without legacy dependencies" as torch itself is a legacy dependency.

9) In addition to obvious statements such as that your model/methodology outperforms other models/methodologies on some/all metrics, could you please comment on cases where it did not (there are quite a few) as understanding is informative.

10) Given how many different tasks/metrics have been employed in this work, could you please comment on if they are the most important tasks/metrics used by all other authors in the area. It was not always clear to me how common, important they are. It is also unclear to me what they do not show. Please comment on limitations of what you are presenting.

**Strengths And Weaknesses:**

Strengths:

- This appears to be a good engineering effort. If you do not have 16x A100 but say only 4 then you could build one in 4 days. With access to just 1 GPU you could get somewhere comparable in 16 days, which is not too bad.
- Although layer-level teacher student is not new, the authors suggest that in this specific context it is. Given popularity of SSLs, this might be of interest to other people working in this area.

Weaknesses:
- The way the manuscript reads is that this is a speech centric work with limited theoretical contribution and largely an engineering effort. In my mind the target audience of this work might lie elsewhere.

---

> ### Author Response · Authors · 2025-09-19
> **Response to reviewer 3tfg (part 1/2)**
>
> We thank the reviewer for their thoughtful suggestions. In addition to the general response above, we address their specific concerns and questions below:
>
> 1. “It appears that you are following https://arxiv.org/pdf/2504.08528 to position your work as "speech language modelling". Without a proper introduction and explanation, I find this terminology/taxonomy of models confusing. Given how much nuances are there and variants to mix speech and language, could you please clearly define what they are and how they are different from models definitely known to the audience of these proceedings.”
>
> We have rewritten the introduction to more directly express the scientific aim of the work. We agree that the terminology around speech language models have been confusing as different authors have used similar terms for completely different objectives and architecture. We have added more context and clarified our aim, which is to learn discrete so-called ‘semantic’ speech units, that encode abstract linguistic information for the purpose of learning a language model.
>
> 2. “Could you please check your manuscript for grammar/semantics. I find phrasing used in many places to be borderline at best. For example, "some approaches added a new step on the units" where I struggle to understand what this mean. Another example is "The selected checkpoint in the one with the lowest validation loss". There may be more than that.”
>
> We have revised the phrasing throughout the manuscript for clarity and correctness.
>
> 3. “I find discussion of terabytes (storage) and weeks (time) to be confusing and somewhat misleading as these units are directly linked with quantities of data used and amount of GPU resource available. Could you please present them with reference to some standard (per hour of speech, per 1xGPU)?”
>
> We have removed the mentions of storage requirements for HuBERT for clarity. We have also added a column containing the total batch size used for pretraining in Table 4. We believe that with this addition, Section 4.5 provides a complete overview of pretraining times and configurations across models.
>
> 4. “One and perhaps the most important claim in this manuscript is that it "makes pretraining significantly more stable and resistant to codebook collapse". I believe you present evidence for codebook collapse albeit it is averaged out in PPL across the data used to compute it. I do not believe I see any evidence presented of the previous methodologies displaying it. I also do not see how you measure stability and how it manifests itself in your experiments. I would also love to see some examples of its lack in existing methodologies. Simply saying that does not suffice.”
>
> Thanks for catching this. We agree that we provided evidence only for codebook collapse, which represents just one facet of stability, one related to the stability of the representation space during training. We have not studied other aspects, such as stability across data domains (acoustic conditions, speakers, etc.) for example. We have removed instances of “stability” in the text, and added potential directions for studying stability in self-supervised speech representation models to the discussion
>
> 5. “Please make it very clear that your methodology requires a certain restriction on the nature of students/teachers (at least as many layers? do you require any other topological and other constraints?)”
>
> We have clarified in the paper that our methodology specifically uses self-distillation, not any kind of student/teacher distillation. Self-distillation does not require pretraining a teacher model with labeled data. Instead, both models are instead jointly using clean and corrupted versions of the input batches. This approach is widely used in self-supervised learning (BYOL, DINO, data2vec). We have added these references to the related work section (paragraph 1).
>
> 6. “You have presented an evidence that pairing all layers of students and teachers is a good idea. Due to model-centric view (everything is a model), I am not fully certain if you have presented your model where only top layers are paired. I do not believe you have presented any evidence of using pairing for some layers only and it is not clear which layers are important to pair and which are not. Could you please explain why none (some) of that investigation has not been done?”
>
> These are very interesting suggestions. We constrained the setting to be close to the original DinoSR setting and keep the same number of parameters and number of prediction heads for maximum comparability. We agree though that further explorations would be very interesting, and now possible, given the acceleration of training time that we provide in our codebase. This is exactly the kind of exploration of the SSL space that we would like to facilitate with our contribution. We have added those considerations to the discussion.

---

> > ### Author Response · Authors · 2025-09-19
> > **Response to reviewer 3tfg (part 2/2)**
> >
> > 7. “I struggle to understand what "using the latest advancements of PyTorch" means.”
> >
> > We have removed this sentence, which originally indicated that our codebase is compatible with torch.compile, resulting in significant performance improvements.
> >
> > 8. “I would be careful making statements such as "hackable foundation for future research without legacy dependencies" as torch itself is a legacy dependency.”
> >
> > We have revised this sentence for clarity. We provide an effective PyTorch-only implementation to pretrain DinoSR and SpidR. This code does not depend on outdated and unmaintained libraries like fairseq for example.
> >
> > 9. “In addition to obvious statements such as that your model/methodology outperforms other models/methodologies on some/all metrics, could you please comment on cases where it did not (there are quite a few) as understanding is informative.”
> >
> > This is a good point; besides the new ASR/SUPERB metrics already discussed, we added a discussion regarding the comparison of ABX and MAP as proxy metrics that could predict the downstream LM metrics (which are our main target metrics).
> >
> > 10. Given how many different tasks/metrics have been employed in this work, could you please comment on if they are the most important tasks/metrics used by all other authors in the area. It was not always clear to me how common, important they are. It is also unclear to me what they do not show. Please comment on limitations of what you are presenting.”
> >
> > Given our reworked introduction we hope it is now clearer that the aim is to measure linguistic comprehension of a language model trained on speech input. The metrics we use are the standard ones used in this area, as defined in the Zero Resource Challenge series (see Dunbar et al 2022 for a review) and subsequent work (eg, Hassid et al 2023): sWUGGY (lexical level), sBLIMP (syntactic level), and tSC. Regarding the metrics for the speech units themselves, the standard metric is ABX, but we add here a comparison with MAP and PNMI as they have also been used as alternative metrics in several publications (Carlin et al, 2011; Hsu et al. 2021). It is actually to our knowledge the first systematic comparison of these metrics across models.

---

### Review · Reviewer_KX3n · 2025-08-11

**Summary Of Contributions:**

The paper proposes SpidR, a self-supervised speech representation model that modifies DinoSR’s architecture by aligning intermediate student layers with corresponding teacher layers for masked prediction. Key contributions include the architectural modification, SpidR uses each student intermediate layer to predict the target instead of using the final layer. According to the experiments, the proposed method improves the training stability and efficiency, and demonstrates gains on zero-shot spoken language modeling(sWUGGY, sBLIMP, tSC) and phonetic discriminability (ABX, PNMI) over HuBERT and DinoSR.

**Audience:**

Yes

**Claims And Evidence:**

Yes

**Requested Changes:**

Include ASR downstream tasks: Evaluate SpidR on standard ASR benchmarks (e.g., LibriSpeech test sets) using frozen features + linear probe or E2E fine-tuning. Compare WERs against HuBERT/DinoSR to validate practical utility.

**Strengths And Weaknesses:**

Strengths:

1、The layer-aligned prediction mechanism is well-motivated (addressing distribution shift in DinoSR) and rigorously validated through codebook stability metrics (Figure 2) and ablation studies (Table 8).

2、The study provides a thorough experimental analysis, systematically investigating scaling laws (Figure 3), rigorously evaluating vocabulary size effects (Table 3), and conducting detailed examinations of layer-wise discriminability (Figures 7, 11-12).

Weakness:

1、Limited novelty: the work offers limited novelty, as its core idea—intermediate-layer prediction—is an incremental extension of DinoSR. While SpidR reorganizes prediction heads (student layer *k* → teacher layer *k*), it retains DinoSR's key components like cross-entropy loss with online codebooks and the teacher-student framework.

2、Insufficient downstream validation: While spoken language modeling (SLM) metrics (sWUGGY/sBLIMP/tSC) are reported, critical speech tasks like ASR are absent. HuBERT are widely used in ASR, yet SpidR’s utility for transcription remains unverified. Without ASR results (e.g., WER on LibriSpeech), the claim of "learning linguistic units" is weakly supported.

---

> ### Author Response · Authors · 2025-09-19
> **Response to reviewer KX3n**
>
> We thank the reviewer for their valuable suggestions and comments. In addition to the general response above, we address their specific concerns and questions below:
>
> - “Insufficient downstream validation: While spoken language modeling (SLM) metrics (sWUGGY/sBLIMP/tSC) are reported, critical speech tasks like ASR are absent. HuBERT are widely used in ASR, yet SpidR’s utility for transcription remains unverified. Without ASR results (e.g., WER on LibriSpeech), the claim of "learning linguistic units" is weakly supported.”
>
> As indicated above, the aim of SpidR is not to improve transcription or build generalist speech features but to learn discrete units that are optimal for ‘pure’ speech-LMs, i.e., language models that are trained without text, from speech only. We would like to argue that this objective is a valid scientific question that can be pursued in its own right without being tied to more traditional text-based downstream applications (e.g., Dunbar et al. 2022).
>
> Given such an objective, results on ASR are only tangentially relevant. ASR requires supervised fine tuning, whereas SLM requires the representations to yield useful discrete units through unsupervised clustering (i.e., without any supervision). In addition, ASR, contrary to SLM requires learning a non trivial P2G model due to irregular pronunciations in English  (/iː/ → ee (see), ea (sea), ie (chief), ei (ceiling), e (me), y (happy)). The requirements for the models are different and it is to be expected that the best models for each type of task will differ. For instance, HuBERT was previously found to be vastly superior to wav2vec 2.0 for SLM (Lakhotia et al, 2021), despite the fact that they are both as much used for ASR.
>
> In order to make this point more explicit, we introduced the evaluation of wav2vec units for SLM in our Table 2, replicating the fact that this model underperforms all of the other SSL models under consideration (including SpidR). We also introduced a new Table 9 in the Appendix with no-LM ASR results, showing that vice versa SpidR underperforms the other SSL models on this task.
>
> - “Include ASR downstream tasks: Evaluate SpidR on standard ASR benchmarks (e.g., LibriSpeech test sets) using frozen features + linear probe or E2E fine-tuning. Compare WERs against HuBERT/DinoSR to validate practical utility.”
>
> As discussed above, we added the ASR results as well as SUPERB phone classification in Table 10. These results illustrate that SpidR brings about a new SOTA for SLM tasks, while losing some ground on the ASR/SUPERB tasks. It remains to be seen whether it is still possible to develop generalist features that would be SOTA at all tasks, or whether this indicates the dawn of specialist SSL systems.

---

> > ### Comment · Reviewer_KX3n · 2025-10-09
> >
> > I understand and accept that the primary aim of SpidR is not to improve ASR.  Your argument that ASR and SLM have different requirements (supervised vs. unsupervised) is valid. However, in the field of speech representation learning, the robustness and generalizability of learned units are typically demonstrated through performance across a suite of tasks. Relying predominantly on a single task (SLM) to claim the superiority of the representations is a narrow approach. The significant performance drop in ASR (as shown in the new Appendix Table 9) is a non-trivial result. It suggests that the discrete units learned by SpidR may be suboptimal for capturing certain phonetic or semantic details that are fundamental to speech understanding, even if they are well-suited for the specific objective of the next token prediction in your SLM setup. A model that excels in one task at a significant cost to others may be learning a biased or overspecialized representation.
> >
> >  I suggest to include results from additional downstream tasks（e. g. SUPERB) that are more aligned with unsupervised representation quality. This will help demonstrate the general utility of the learned units.

---

> ### Author Response · Authors · 2025-10-28
> **Response to Reviewer KX3n**
>
> We agree with the distinction raised by the reviewer between general purpose units and specialised units. Where we diverge, is that we maintain that it is a valid research question to work on specialised units, and in particular specialised linguistic units for downstream SLM training. The unsupervised discovery of linguistic units has been a long standing objective in the speech community [1,2,3] and has recently made progress in several speech modeling research papers [4,5,6,7,8].
>
> Therefore, we would like to maintain that working on these specialized units is a perfectly valid research area, distinct from the equally valid research area of generic units that can be fine tuned for specific tasks, as promoted in the SUPERB benchmark.
>
> The difference between types of speech units across a variety of tasks is itself a fascinating topic worth further research, but an exhaustive comparison across units and tasks would go beyond the scope of this paper. We propose nevertheless to introduce additional analysis with [this figure](https://drive.google.com/file/d/1-Offbx_iX0drcwnLDnGKTrr-h_BG9xku/view?usp=sharing), comprising results from tables 2 and 9, which shows the absence of positive correlation between SLM performance and ASR or PR performance which we interpret in terms of the different requirements of these different tasks. ASR and PR are supervised tasks that can benefit from information spread across different layers, and can easily ignore irrelevant information (like speaker, noise, etc) through use of classifier heads. SLM metrics in contrast require the models to concentrate the phonetic information in a single layer, and reduce the irrelevant dimension to a low enough variance subspace, such that unsupervised clustering ignores it. Metrics such as ABX and PNMI which measure the direct access of phonetic information in the embeddings in an unsupervised fashion therefore correlate more strongly with SLM metrics (as shown in Figure [14] in the paper) and are more relevant for our work than SUPERB tasks.
>
> We propose to introduce this in the appendix, and thank the reviewer for highlighting the distinction between general-purpose units and specialised units. We hope that this will answer the reviewer's concerns, and we remain open to adding more results to strengthen our SLM evaluation (e.g., alternative SLM setups).
>
> [1] Versteegh, M., Thiollière, R., Schatz, T., Cao, X.N., Anguera, X., Jansen, A., Dupoux, E. (2015) The zero resource speech challenge 2015. Proc. Interspeech 2015, 3169-3173, doi: 10.21437/Interspeech.2015-638
>
> [2] E. Dunbar, N. Hamilakis and E. Dupoux, "Self-Supervised Language Learning From Raw Audio: Lessons From the Zero Resource Speech Challenge," in IEEE Journal of Selected Topics in Signal Processing, vol. 16, no. 6, pp. 1211-1226, Oct. 2022, doi: 10.1109/JSTSP.2022.3206084.t
>
> [3] Kamper et al., “Unsupervised Word Segmentation and Lexicon Discovery Using Acoustic Word Embeddings” in IEEE/ACM Transactions on Audio, Speech, and Language Processing (2016)
>
> [4] On Generative Spoken Language Modeling from Raw Audio https://aclanthology.org/2021.tacl-1.79 (Lakhotia et al., TACL 2021)
>
> [5] Z. Borsos et al., "AudioLM: A Language Modeling Approach to Audio Generation," in IEEE/ACM Transactions on Audio, Speech, and Language Processing, vol. 31, pp. 2523-2533, 2023, doi: 10.1109/TASLP.2023.3288409.
>
> [6] Baade et al., “SyllableLM: Learning Coarse Semantic Units for Speech Language Models”, ICLR 2025
>
> [7] Chang, H.-J., Gong, H., Wang, C., Glass, J., Chung, Y.-A. (2025) DC-Spin: A Speaker-invariant Speech Tokenizer for Spoken Language Models. Proc. Interspeech 2025, 5723-5727, doi: 10.21437/Interspeech.2025-246
>
> [8] Chou et al., “Flow-SLM: Joint Learning of Linguistic and Acoustic Information for Spoken Language Modeling” (2025), https://arxiv.org/abs/2508.09350

---

> > ### Comment · Reviewer_KX3n · 2025-10-29
> >
> > Thank you for your detailed response and for agreeing to include the additional analysis in the appendix. The provided figure and explanation effectively address my concern regarding the distinction between general-purpose and specialized units. I am satisfied with the proposed changes and consider my concern resolved.
> >
> > I recommend acceptance of the paper.

---

### Review · Reviewer_nrt3 · 2025-09-08

**Summary Of Contributions:**

The paper proposes SpidR, a self-supervised method for learning speech representations based on an improvement of the previously proposed DinoSR. The key idea is a modification of the DinoSR framework, based on taking advantage of the student’s own intermediate representation. Compared to DinoSR, SpidR turned out to be more stable and to better capture phonetic information when its learned units are used in spoken language modeling. The authors also focus on training efficiency, showing that SpidR can be trained with reasonable computational resources, making it more accessible.

**Audience:**

Yes

**Broader Impact Concerns:**

No ethical concerns for this work.

**Claims And Evidence:**

No

**Requested Changes:**

Based on the weaknesses I noticed, I would recommend that the authors consider the following changes:

* Include metrics beyond phonetic consistency (e.g., acoustic consisentcy, gender consistency, emotion consistency, etc). If not feasible, provide a clear justification.
* Assess the performance on different downstream tasks, such as those in SUPERB or similar benchmarks. While I understand it may not be feasible to cover all downstream tasks, even one or two additional evaluations would strengthen the paper. If not applicable or relevant, explain why.
* Compare the proposed method with popular audio codes (e.g., Encodec, Mimi). If not relevant, justify why.
* Include a comparison with WavLM, or explain why the it is omitted, as WavLM often outperforms HuBERT on many SUPERB tasks.
* Report the scaling laws for DinoSR.
* Expand the Self-Supervised Speech Representation Learning section with additional works from the literature.

**Strengths And Weaknesses:**

Strengths:

* The paper is well-written, and the proposed method is explained clearly and rigorously. The key differences from DinoSR are highlighted properly.
* While the technique is a simple modification of DinoSR, the performance improvements are significant, making this a valuable contribution.
* The authors worked on an efficient implementation, allowing the model to be trained with reasonable computational resources.

Weaknesses:

* Limited Evaluation: The evaluation primarily focuses on phonetic content and consistency. Speech, however, can capture richer features such as emotion, accent, prosody, and speaker identity, which are not fully considered here. By focusing mostly on phonemes, I think we are not fully evaluating the potentialities of spoken language models. A broader evaluation, including acoustic, gender, and emotion consistency, would make the results of the paper more valuable for the community.
* Downstream tasks: Connected to the point above, it looks like the authors do not report results on popular downstream benchmarks (e.g., SUPERB). While DinoSR is also limited in this aspect, at least it reports some results for ASR. I think that, while evaluating spoken language modeling is totally fine, assessing performance on downstream tasks can provide deeper insights into the actual model performance.
* Comparison with audio tokens: There is no comparison with modern audio token methods (e.g., Encodec, Mimi), which I think are relevant alternatives to the proposed approach.
* WavLM: Hubert is a strong baseline, but WavLM has been shown to outperform it in many tasks (see the SUPERB Benchmark). I would have expected a comparison with WavLM.
* Scaling laws: Figure 3 and Table 3 do not report scaling laws for DinoSR. It is not clear why to me.
* Related work coverage: The Self-Supervised Speech Representation Learning section could be broader, including additional relevant works such as WavLM, BesRQ, and others.

---

> ### Author Response · Authors · 2025-09-19
> **Response to reviewer nrt3**
>
> We thank the reviewer for their valuable suggestions and comments. In addition to the general response above, we address their specific concerns and questions below:
>
> - “Include metrics beyond phonetic consistency (e.g., acoustic consisentcy, gender consistency, emotion consistency, etc). If not feasible, provide a clear justification.”
>
> As explained in the general response, we focus on the spoken language modeling task. Success requires the model to learn representations that can be discretized into meaningful tokens, corresponding to sub-linguistic units. Therefore, the most salient information in the model’s representations should be linguistic (phonemic or word-level), not acoustic or speaker-related. While we acknowledge that acoustic and speaker consistency are essential for building systems that interact with humans, they fall outside our current scope. Previous spoken language modeling work tried to improve resynthesis fidelity and speech continuation quality by incorporating prosody (pGSLM) or expressivity (Spirit LM) through additional F0 or style tokens. We have added those considerations to the discussion.
>
> - “Assess the performance on different downstream tasks, such as those in SUPERB or similar benchmarks. While I understand it may not be feasible to cover all downstream tasks, even one or two additional evaluations would strengthen the paper. If not applicable or relevant, explain why.”
>
> As explained above, performance of our encoder in downstream ASR is not the target of our paper. But for comparability with other speech encoders, we have added ASR evaluation with finetuning from the Libri-Light limited, and ASR and phone recognition from SUPERB using frozen features in Appendix D. As expected the ASR performance of SpidR is below the current SOTA (although on par with popular models like wav2vec 2.0). Interestingly, even though SpidR units are better than DinoSR units for training LMs, the reverse is true for SUPERB phoneme classification. We believe that this is because supervised classification diverges from unsupervised clustering (typically used for spoken language modeling) due to the existence of subspaces dedicated to different types of information within the embeddings (Liu et al., 2023). We discuss these results in the Appendix.
>
> - “Compare the proposed method with popular audio codes (e.g., Encodec, Mimi). If not relevant, justify why.”
>
> As discussed above different audio encoders have been developed with different purposes in mind. Audio codec are developed for high fidelity audio compression (and trained with a reconstruction loss in audio space), and are typically worse than models like HuBERT or SpidR (trained with a masked objective in latent space) when it comes to provide semantic units for modeling spoken content, as demonstrated in Table 10 of https://openreview.net/forum?id=eqNchtvc6v. The Mimi encoder attempts to combine both types of codes by distilling WavLM into Encodec units. Other approaches, like in AudioLM use a cascade of “acoustic” tokens from a codec (SoundStream) and “semantic” tokens from w2v-BERT. In this paper, we are concerned with improving the semantic component of such composite models.
>
> - “Include a comparison with WavLM, or explain why the it is omitted, as WavLM often outperforms HuBERT on many SUPERB tasks.”
>
> This is a good point; we only included ‘base’ SSL models that are trained from scratch on the same dataset (Librispeech), and only differ in architecture and training loss. WavLM belongs to a class of derived SSL models that fine-tune existing SSL models by adding a noise resistance objective through data augmentation and are typically trained in multiple stages (other such methods include Spin, R-Spin, and DC-Spin).
>
> In this paper we provide and extensively test a new base model, SpidR, and leave for future work the exploration of noise resistance with this model. We still include the comparison with WavLM in Table 2, with a note explaining the difference between HuBERT and WavLM.  We find that despite the fact that SpidR is trained without additional noise resistance objectives, its k-means units outperform WavLM on downstream spoken language modeling.
>
> - “Report the scaling laws for DinoSR.”
>
> Thank you for this valuable suggestion. We are currently running these experiments and will update the figure once results are available, before the end of the discussion period.
>
> - “Expand the Self-Supervised Speech Representation Learning section with additional works from the literature.”
>
> We have highlighted the existing WavLM citation and added a reference to BEST-RQ.

---

> ### Author Response · Authors · 2025-09-23
> **Response to reviewer nrt3**
>
> We have added the scaling laws for DinoSR in Figure 15 in the appendix. Trends remain consistent across scales for models evaluated--SpidR performs best on spoken language modeling metrics, followed by HuBERT, followed by DinoSR.

---

> > ### Comment · Reviewer_nrt3 · 2025-10-31
> >
> > I would like to thank the authors for their efforts in addressing my previous comments as well as those of the other reviewers. The manuscript has clearly improved since the first round of reviews.
> >
> > While the results presented are potentially interesting to community, I believe the impact of this work could be significantly better by going beyond the focus on semantic units. Speech is a rich and complex signal that conveys not only linguistic information but also paralinguistic cues such as emotion, speaker identity, and prosody, which play an essential role in many modern applications. Moreover, in the context of multimodal large language models, it is increasingly important to evaluate whether the proposed tokens can be effectively used for generative tasks, not just for recognition or classification. These aspects are not explored in the current version of the paper.
> >
> > In my opinion, current research should aim to develop more universal and flexible tokenizers that can serve a wide range of downstream applications, rather than focusing exclusively on tokenizers optimized for a specific type of task (e.g., semantic tokens).
> >
> > That said, I would not oppose acceptance if the majority of the reviewers and the editor are in favor of it.

---

> ### Author Response · Authors · 2025-11-05
>
> We thank the reviewer for their comment and agree that the next step would be to build encoders that not only learn “semantic units”, but learn disentangled representations for complementary aspects of the speech signal: semantic units, prosodic units (f0, energy, duration), expressive units (whispered, shouted, angry, sad, etc), and speaker units. It would be interesting to show that learning these representations jointly in a single encoder could actually improve over learning each of them separately, although disentanglement within a purely self-supervised approach could be challenging [1-5]. We propose to add this perspective and these references in the general discussion.
>
> [1] Qian et al. Contentvec: An improved self-supervised speech representation by disentangling speakers. ICML 2022
>
> [2] Kharitonov et al. (2022). Text-Free Prosody-Aware Generative Spoken Language Modeling. ACL 2022.
>
> [3] Polyak et al. (2021) Speech Resynthesis from Discrete Disentangled Self-Supervised Representations. Interspeech 2021.
>
> [4] Lin et al. (2023). "Self-supervised neural factor analysis for disentangling utterance-level speech representations." ICML 2023
>
> [5] Tu et al. (2024) "Contrastive self-supervised speaker embedding with sequential disentanglement." IEEE/ACM Transactions on Audio, Speech, and Language Processing

---

### Author Response · Authors · 2025-09-19
**General response**

Among the detailed comments of the three reviewers, a common theme concerns the downstream evaluations of the models, either asking for the importance of these metrics (Rev 3tfg) or asking to add new ones (Rev. nrt3 and KX3n). We thank the reviewers for these useful remarks which highlight a shortcoming in the previous version of the paper regarding the explanation of the scientific aim of this work. We propose to address this shortcoming through better introduction, related work and discussion, in addition with new results in the Appendix section.

In more details, speech features have been developed to serve diverse purposes (speech compression, semantic units extraction, expressive modeling, ASR, and more). Even though benchmarks like SUPERB encourage to develop units that would be good for a variety of tasks, we often find that different objectives naturally demand different optimal features, a principle already demonstrated in various contexts (see https://aclanthology.org/2021.tacl-1.79/ which finds that HuBERT outperforms wav2vec 2.0 for spoken language modeling even though they are typically on par at ASR (cite HuBERT), and Table 10 of https://openreview.net/forum?id=eqNchtvc6v which finds that SSL models perform better than codec models on spoken language modeling but worse on acoustic modeling)

In this context, our specific aim in this paper is to focus on developing “semantic” speech units for the purpose of spoken language modeling, an area we believe is critically important and understudied. We motivate this objective in the introduction by drawing inspiration from children who learn a spoken language before learning to read and write, and therefore are able to learn a language model from speech inputs only. The main direction of research in this area has historically been to find discrete speech units, which may or may not align with phonemes, but are able to sustain language learning through use in an LM in place of text (Lakhotia et al, 2021). This line of investigation typically evaluates the knowledge learned by the speech language model at various linguistic levels (phonetic, lexical, syntactic semantic) with zero-shot metrics inspired by human psycholinguistic methods as developed in the zero resource challenge series (Dunbar et al, 2022) (hence the ABX and grammaticality judgments methods). One additional contribution of this paper is in fact to demonstrate that metrics at the phonetic level (measured at the output of the speech encoder) can function as proxies that predict the metrics in the downstream LM (see Figures 13 and 14).

In this context, the metrics used for the other use cases of speech units (ASR, compression, expressive modeling, etc) become less directly relevant. We nevertheless include some of them in the Appendix for reference purposes, as we believe this could be useful for a broader understanding of which speech features are good for which task

We have uploaded a revised version of the paper with changes highlighted in red. The main modifications include:

1. Improved clarity of scientific goals: We have incorporated reviewer feedback to improve our introduction and related work sections, providing a more detailed overview of spoken language modeling, and clearly specifying what falls within or outside our scope. We highlight how model choices differ across tasks of interest.
2. Additional experiments:  Following requests from two reviewers, we have added ASR and phone recognition in Appendix D. These experiments use default hyperparameters from wav2vec 2.0 Libri-Light finetuning or SUPERB benchmark without further optimization.
3. Expanded comparisons: We have added comparisons to Wav2vec base and WavLM for spoken language modeling, demonstrating improved performance over both models for spoken language modeling metrics. We are currently running scaling analyses for DinoSR. We have updated the codebase for training spoken LMs, which results in slightly different scores, though the overall trends remain unchanged.

We hope that the reframing of the objective of the paper and specific modification will address both the main and detailed concerns of the reviewers.

---

### Comment · Action_Editor_NGMv · 2025-10-06
**Engage into discussion**

Dear Reviewers and Authors,

As we extended discussion period, please engage into discussion to see if the raised concerns can be resolved or what concerns still remain and why. Also as a reminder, TMLR doesn't require novelty, the key things to focus on are interesting results to the community and correctness of the theoretical and empirical validation of the ideas.

Thanks,

AE.

---

### Decision · Action_Editor_NGMv · 2025-11-25

**Recommendation:** Accept with minor revision

**Additional Comments:**

I did a pass over the final paper revision as well as all comments from reviewers. I would like to point the following pieces which I request authors to fix for the camera-ready paper:

- Abstract should be expanded and be more specific on terms usage such as "speech representations" (reflect semantic information), "spoken language modeling" (reflect it is for textless speech models), "downstream language modeling metrics" (specify what metrics), "previous state-of-the-art methods" (specify which models are they). Abstract does not reflect 3 main contributions in the current state, and I believe it should also reflect exactly the clear task you are targeting and property of the tokens (given long discussion with reviewers).
- SSL related works you need to start with the clear sentence what these SSL all prior works were built for (general representations to solve many speech tasks at the same time, also specifically over-optimized for ASR -- see e.g. HUBERT original paper how all hyperparameters were optimized), and that it is different usage now for these representations as people use them to train speech LLMs (both textless and speech-text based). Right now I see a clear picture from the introduction, but your work is not put / compared to related works: in the SSL subsection it seems that you introduce better SSL / more efficient SSL but the original context for SSL is different downstream tasks which people often optimize, thus it creates confusion.
- Please add examples of model generation, so that we can see what model is capable of generating. (I assume here that you have a decoder to restore the speech wave back from the speech tokens. Correct?)
- Please add a note why OPT model is used instead of say GPT-style arch, and why you focus only on one model size, not bigger one e.g. having 300M similar to prior works having both base and large configurations. This could also explain why e.g. scaling saturates with data size -- I suspect bigger models will be needed.
- On scaling law -- it feels that you improved the model in general over HUBERT tokens but scaling itself is not improved as curve slope seems to be the same. Maybe adding some comments on that will be helpful.
- I think your work and ablations are critical and will have a long impact based on the prior and your observations that better SSL does not lead to better tokens for SLMs. I think broader discussion at the end of the paper will be super helpful for the community -- discuss how people actually focused for a long time on ASR improvements from SSL representations, then added SUPERB but still models were heavily optimized for ASR. Moreover, current tokenizers for speech LLMs are using ASR as auxiliary task or they use ASR representations (from whisper) directly to plugin into LLM -- this may be a wrong direction based on your results. Also we may want to test other metrics again, like ABX or other to find stronger correlation between metrics we measure and speech LLMs performance.
- I understand why you compare mainly with HUBERT tokens as they are heavily used in all speech LLMs work (primarily as the model was OSS compared to other many SSL works). But why do you not compare with RVQ tokens (speechtokenizer) taking only the first channel of it which corresponds to semantic? Could you add some discussion around them to clarify this point? I think this comparison is really needed to unify the community between acoustic + semantic tokens vs semantic tokens: we need to have a comparison of how semantic representation from "acoustic + semantic tokens" behaves.
- Please include detailed description of metrics and finalize introduction to make the paper more accessible to the audience out of speech domain.

If you have any questions regarding changes I request feel free to reach out !

**Audience:**

Yes

**Audience Explanation:**

Reviewer 3tfg expressed issues that the paper is a very speech centric work, and thus interest may be limited. Also Reviewer is concerned about metrics used, that many TMLR readers may not be aware of. Other reviewers did not raise any issues with respect to the audience.

**As AE I would like to comment on that issue, and raise personal concerns on such perception other people may have:**
- First, I believe the current paper falls into the scope and interest of TMLR, as TMLR covers broad topics in ML, there are plenty of speech-related papers that were submitted, published, and many reviewers and AEs are from the speech / audio domain.
- I am really happy to see that TMLR is a place where the speech community can publish too, as many speech-oriented conferences are not suited for long papers, in depth investigations and analysis. Moreover, submitting and publishing in a broader topic places is a "must practice" as otherwise general ML is not developed to solve broader tasks.
- The problems on speech tokenization and how these tokens can be incorporated into text-speech models are of high priority in the community in general: this is related to many aspects like what semantic and non-semantic information we need to learn, how to align things between different domains, etc. These findings can help for vision-language tasks, for video-text modeling and general multi-modal models. Besides, speech is the way we communicate!

With this, I believe, the paper is strongly aligned with the TMLR audience, and will be of interest to other domains too. At the same time I agree that better introduction and description of metrics and problems could be done to attract more people and make the paper more accessible to a wider audience. Thus I request authors to make some changes for the camera ready to improve accessibility to a wider audience.

**Claims And Evidence:**

Yes

**Claims Explanation:**

Nowadays, to build speech LLMs, the community uses LLM machinery and thus builds discrete tokens to represent speech. These discrete tokens often come from a combination of acoustic tokens (built e.g. via reconstruction loss) and semantic tokens (built e.g. via clustering SSL representations). Particularly, the scientific question is can we build textless models similar to the way babies are learning language? Authors in the paper consider textless speech LM building using semantic tokens. Authors propose an improved SSL training that 1) reduce compute to get strong representations; 2) representations are useful for and target specifically textless speech LMs as a downstream task (e.g. authors show that these representations are not strong for ASR or some other speech-related tasks); 3) proposed SSL outperforms prior works like HUBERT and DinoSR for textless speech LMs.

All reviewers in the end agreed that **the paper has rigorous analysis, the method is well motivated, and the efficiency of the method brought by authors due to engineering efforts is useful for the community.**

**The key points where reviewers are concerned for the final decision are:**
- speech-centric topic (raised by Reviewer 3tfg)
- novelty (raised by Reviewers KX3n, nrt3)
- necessity of broader evaluation of representations (include more tasks and tokenizers to compare on and with) (raised by Reviewers KX3n, nrt3)

*Let me comment on every of these points separately:*
- Speech-centric topic
  - This is discussed in details below under "Would at least some individuals in TMLR's audience be interested in knowing the findings of this paper?"
- Novelty
  - Based on the TMLR policy I set aside this point in decision making.
- Broader evaluation
  - Reviewer KX3n is satisfied by added comparisons and changes in the final revision.
  - Authors reworked introduction and related works and (in my view) well-motivated why they do not compare with acoustic tokenizers as well as SUPERB benchmark. Authors also add discussion and results that SSL representations strong for ASR do not necessarily work for spoken LMs.
  - I agree with the Reviewer 3tfg who was pointing in the final recommendation that it has nothing to do with HUBERT and SUPERB original design and the paper targets different downstream tasks. The paper even considers different metrics on how to evaluate and select the representations to be used for spoken LMs compared to HUBERT.

I think the remaining debatable point (raised in the final recommendation by Reviewer nrt3) is about how useful to work on semantic tokens only: should we target developing / pushing semantic tokens only or focus on general tokens development which solve both acoustic quality of generation and semantic language modeling (and thus comparison with e.g. RVQ tokens is needed)? The paper is focusing only on the second aspect (semantic) and in the end may not be generalizable to solve the problem we have (building speech LM with both language understanding and high fidelity speech generation abilities; supporting multimodal modeling). **Given current state of research where we don't have consensus on speech tokens, the best recipe and downstream tasks we are targeting I believe this debate should continue (which is great to see e.g. in the current TMLR submission discussion) while the paper in the current form (due to wide usage of HUBERT tokens as semantic in speech LLM research) gives helpful insights which will be helpful to resolve this debate.**